# *Pseudomonas aeruginosa* dynamically prioritizes motility and resource recycling during prolonged starvation

Findlay D. Munro,[1] Elize Ambulte,[1] Claudia M. Hemsley,[1] Megan Bergkessel[1]

**ABSTRACT**  Heterotrophic bacteria rapidly deplete essential macronutrients during growth and must navigate subsequent periods of growth arrest imposed by starvation. Nutrient limitations can be dynamic in nature, requiring ongoing regulatory adjustments involving new protein synthesis despite total biosynthetic activities being dramatically lower than during growth. Here, we have characterized the responses of the opportunistic pathogen *Pseudomonas aeruginosa* to prolonged starvation for carbon or nitrogen sources and to transitions between these states. We find that most cells survive both types of starvation for more than a week and maintain low but robustly detectable levels of protein synthesis in the absence of growth. Nitrogen-starved cells are larger, make more proteins, and retain fewer ribosomes than carbon-starved cells, indicating that distinct physiological strategies are adopted during the two starvation types. We found that the newly synthesized proteomes of each starvation type are distinct although many of the most highly synthesized proteins are shared between both conditions. Interestingly, we observed a temporary burst of protein synthesis as cells were transitioned between the two starvation conditions, which may reflect active remodeling of the proteome during growth arrest. We also used transposon insertion sequencing to identify genes impacting fitness in both starvation conditions and during transitions between the two and found that a highly overlapping set of global regulators most strongly influenced survival. Combining these data sets, we highlight proteases and chaperones, flagellar motility, and the nitrogen-related phosphotransferase system as key fitness-impacting functions that are actively maintained by growth-arrested *Pseudomonas aeruginosa*.

**IMPORTANCE**  Molecular microbiology has traditionally focused on exponential growth in model organisms as the preferred context in which to study bacterial physiology, especially the regulation of new protein synthesis. However, in natural environments, including many infection contexts, bacteria frequently enter growth arrest due to nutrient limitation. The dynamics and regulation of protein synthesis in growth-arrested cells remain poorly understood, especially in pathogens. Furthermore, growth arrest increases tolerance to a variety of stresses, including many clinically used antimicrobials. We have conducted a comprehensive exploration of the proteins being made by growth arrested *Pseudomonas aeruginosa* during total nitrogen or carbon starvation and at the transition between these two starvation types, and the genes supporting fitness under these conditions. These datasets suggest dynamic redistribution of resources among important cellular functions and will serve as a resource for further investigations of starvation-induced growth arrest, a ubiquitous but understudied physiological state of heterotrophic bacteria.

**KEYWORDS**  *Pseudomonas aeruginosa*, microbial dormancy, carbon starvation, nitrogen starvation, BONCAT, proteomics, Tn-Seq

Address correspondence to Megan Bergkessel, mbergkessel001@dundee.ac.uk.

The authors declare no conflict of interest.

See the funding table on p. 20.

10.1128/msystems.01439-25  **1**

Historically, bacterial physiology and regulation have often been studied under conditions that support balanced, steady-state, exponential growth. These efforts have permitted quantification of the relationships between growth rate, ribosome abundance, translation rate, and nutrient quality (the "bacterial growth laws," reviewed in references 1, 2 and references therein) and have provided essential insights into the coordination of metabolism with gene expression during rapid growth. For example, concentrations of proteins in the *E. coli* proteome are determined almost entirely by transcription initiation rates at their promoters (3), and the fraction of the cell's mass that is comprised of ribosomes is linearly related to the growth rate (4). Rapid, steady-state growth also offers practical advantages for studying responses to perturbations. During rapid growth, high rates of metabolic activity, gene expression, and proteome turn-over can power robust and easily measured responses.

However, many observations from natural environments suggest that bacteria frequently exist in physiological states outside of steady-state growth. Bacteria that grow in association with a host are often constrained by the host, leading to growth rate estimates that are sometimes surprisingly slow or highly variable (5–7). Population blooms of organisms that were scarce before nutrient amendment suggest the presence of small dormant populations of many species in complex microbiomes (8). Even in nutrient-rich conditions such as laboratory LB batch cultures, steady-state growth is fleeting, ending at optical densities ($OD_{600}$) near 0.1, because nutrients are rapidly depleted by growing heterotrophic bacteria (9).

When colonizing a finite niche, the depletion of essential nutrients is a common reason for bacteria to exit steady state growth. While the initial exit from exponential growth into stationary phase has been well-characterized in laboratory cultures, at least in *E. coli* (10–12) ongoing biosynthesis and its regulation during protracted starvation is less explored. In part, this is historically due to technical challenges. Although imposing starvation for an essential nutrient in the laboratory can be straightforward, growth-arrested cells severely reduce rates of key activities. Important regulatory changes to new protein synthesis can, therefore, have small and potentially undetectable impacts on measured total protein levels. Furthermore, diverse studies have suggested that population heterogeneity may increase in growth arrest (13, 14), such that the low average protein synthesis seen in these populations may result from a subset of the population that remains more active while other cells enter dormancy. Studying the low-level and heterogeneous activity maintained by growth-arrested bacteria, therefore, requires highly sensitive methods to measure per-cell activity.

Fortunately, the development of methods for fluorescently labeling proteins, RNAs, and metabolites in single cells offers insight into the distribution of per-cell activity across non-growing populations. BONCAT (bio-orthogonal non-canonical amino acid tagging) is a powerful method to mark proteins newly synthesized during a specific period of labeling via incorporation of a click chemistry-compatible non-canonical amino acid (azidohomoalanine, AHA) by the native translational machinery (15, 16). BONCAT permits the estimation of total new protein synthesis per cell if the AHA in fixed cells is conjugated to a fluorescent dye (17) and can also allow for enrichment and identification by LC-MS/MS of the nascent proteins made by the whole population of cells if lysate is conjugated with functionalized agarose beads (18). Together, these methods can reveal how low levels of activity are distributed across growth-arrested populations and identify molecular players that comprise that activity.

A second obstacle to the study of growth arrest physiology and regulation is conceptual: in the absence of growth, the functions that should be prioritized for maximizing fitness are not immediately obvious. Previous work has suggested that even among Gammaproteobacteria, starvation survival strategies may be distinct, with some organisms retaining higher rates of new protein synthesis than others during protracted carbon starvation (17). In many bacteria, starvation survival strategies include stockpiling non-limiting resources within high-density stores, such as carbon-rich polyhydroxyalkanoate (PHA) granules (19) or nitrogen-rich hibernating ribosomes (20, 21). Methods to

evaluate mutant fitness, such as Tn-Seq/TRADIS, have permitted the identification of functionally important genes in diverse growth arrest conditions, allowing new insights into regulatory strategies that contribute to starvation survival (22, 23). By combining all these approaches, we hope to gain insight into the key functions that are preferentially maintained under severe resource limitation.

In this work, we have focused on the starvation responses of *Pseudomonas aeruginosa,* an opportunistic pathogen that has been shown to enter starved states in biofilms and other infection contexts (24–26). After growing cultures into stationary phase in rich media (lysogeny broth [LB]), we switched planktonic cells to either nitrogen source-free or carbon source-free defined minimal media at a moderately low density for 24–200 h as model growth-arrested conditions. This method of inducing starvation was chosen to allow cells to gradually adapt to slowing growth among multiple carbon sources, which has been shown to promote starvation survival (27, 28), but avoid stressful changes in pH or oxygen availability associated with death in prolonged high-density LB stationary phase (17, 29). Carbon starvation is expected to impose limitation for both energy and biosynthetic substrates (nucleotides and amino acids), while nitrogen starvation limits the availability of biosynthetic substrates but does not directly limit energy. In the absence of growth, there are no obvious constraints relating biosynthetic activity to cell size or ribosome abundance, so we have measured population size, culture optical density (at 500 nm to avoid potential absorbance by the phenazine pyocyanin at the more typical 600 nm wavelength), cell size, motility, ribosome abundance, and new protein biosynthesis to gain insight into global physiological parameters in these two different starvation conditions. We have also measured the total proteomes, newly synthesized (nascent) proteomes, and genes impacting fitness using standard label-free proteomics, BONCAT proteomics, and Tn-Seq. Finally, we investigated new protein synthesis and fitness determinants during transitions between starvations to explore conditions where remodeling the proteome might be beneficial despite ongoing growth inhibition.

We highlight three functional categories of genes that are relatively highly expressed during starvation and impact fitness: flagellar genes, proteases and chaperones, and the nitrogen-related PTS system (PTS$^{Ntr}$). We show that mutations in representative genes from these categories cause diverse perturbations to total protein synthesis, ribosome abundance, morphology, and motility during growth arrest. When considering all our data, we propose that starved *P. aeruginosa* can store non-limiting nutrients and dynamically redistribute these resources to power key activities. Ultimately, we hope that these data sets can serve as a resource for further explorations of physiology in starvation-induced growth arrest.

## RESULTS

### Defining the translational capacity of growth-arrested *P. aeruginosa*

We first broadly characterized the response of *P. aeruginosa* to complete starvation for nitrogen or carbon sources. Overnight *P. aeruginosa* LB cultures were pelleted and resuspended at low density (OD$_{500}$ 0.2) in MOPS-buffered minimal media lacking either ammonium chloride (nitrogen source) or sodium succinate (carbon source) and were sampled across several days. As expected, the absorbance of starved cultures did not measurably increase over 10 h (Fig. S1A), suggesting that cells cannot use the MOPS buffer as a carbon or nitrogen source and did not grow under either starvation condition. However, clear distinctions in culture absorbance, CFU counts, and cell area became apparent when starved cultures were tracked for prolonged periods. Carbon-starved cultures slightly decreased in absorbance, CFU counts, and average cell area (as a measure of cell size) especially at later timepoints (Fig. 1A through C), suggesting a gradual loss of biomass. Nitrogen-starved cultures showed more dynamic trends, increasing in CFU counts during the first 8 h of starvation before stabilizing (Fig. 1A). The absorbance of nitrogen-starved cultures almost doubled over 100 h of incubation (Fig. 1B) and the average area of nitrogen-starved cells quickly decreased following

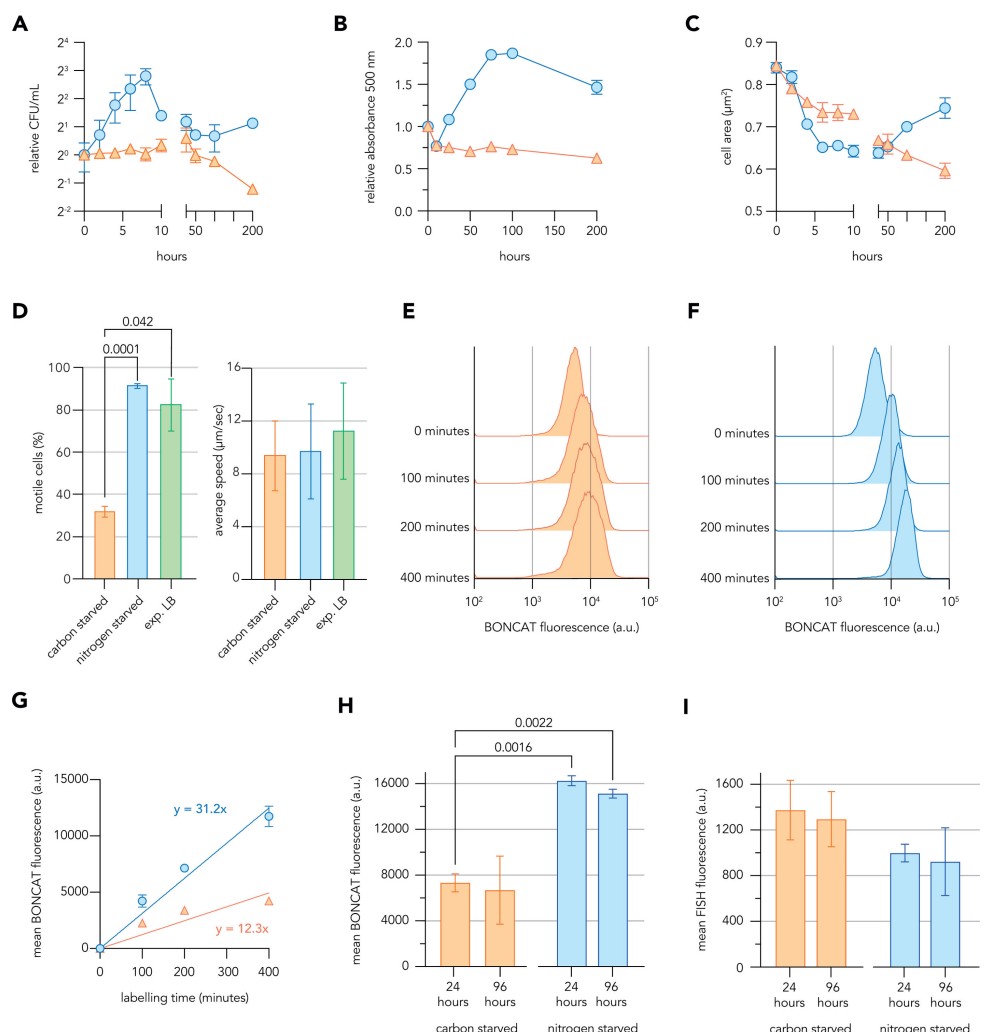

FIG 1 Defining the protein synthetic capacity of growth-arrested *P. aeruginosa*. (A to C) Longitudinal trends in colony-forming units (A), absorbance 500 nm (B), and average cell area (C) of *P. aeruginosa* cultures following resuspension in nitrogen starvation minimal media (blue circles) or carbon starvation minimal media (orange triangles). Note discontinuous *x*-axes for panels A and C intended to show detailed dynamics during the first 10 h of starvation. Cell area measurements were derived from the microscopy of samples fixed at indicated timepoints. (D) Bar graphs show the percentage of motile cells and the average speed of motile cells quantified from videos of *P. aeruginosa* cultures sampled following 48 h of starvation for carbon or nitrogen in minimal media or growing exponentially in LB rich medium. (E and F) Histograms show the distributions of new protein synthesis levels as per-cell fluorescence values in *P. aeruginosa* cultures starved of carbon (E) or nitrogen (F) in minimal media for 48 h and BONCAT labeled with 500 µM AHA for time periods indicated. (G) Line graph shows the relationship between BONCAT labeling time and average fluorescence of *P. aeruginosa* cells starved of carbon (orange triangles) or nitrogen (blue circles) in minimal media for 48 h, with the gradient of each line shown. (H and I) Mean fluorescence values of *P. aeruginosa* cells starved of carbon or nitrogen in minimal media for 24 or 96 h and BONCAT labeled for 240 min with 200 µM AHA (H) or FISH stained with ribosome-directed probes (I). Summarized data represent the mean values calculated from three biological replicates with error bars displaying standard deviation. Histograms show data from a single representative biological replicate. One-way analysis of variance was conducted to assess differences among conditions and time points, with Dunnett's T3 multiple comparisons test used to calculate adjusted *P*-values for pairwise comparisons and significant values shown. a.u. = arbitrary units.

resuspension in starvation media before increasing with prolonged incubation (Fig. 1C). These trends could be explained by initial reductive divisions upon resuspension in nitrogen starvation media, as has been observed in *E. coli* (30), followed by subsequent changes in cell morphology and culture opacity that occur independently of

cell divisions. Interestingly, motile cells were observed in both nitrogen-starved and carbon-starved cultures (Fig. 1D), suggesting that flagellar motility is maintained by sub-populations of non-growing *P. aeruginosa* during days of nutritional deprivation, an observation also recently reported by others (31). While carbon starvation significantly reduced the motile fraction of the population (defined as cells moving more than 4 µmwhile tracked, a threshold determined by investigating flagellar mutant *flgE::tn* cells, Fig. S1D), the motile fractions were similar during nitrogen starvation and LB growth, and the motile cells swam at similar speeds in all three conditions. In general, *P. aeruginosa* showed dynamic responses to the distinct starvations and, following an initial ~10 h adaptive phase, cells appeared to enter a prolonged growth-arrested state with stable population numbers in both starvation conditions.

We then probed the translational capacity of growth-arrested *P. aeruginosa* under both starvation conditions. BONCAT was used as a proxy to measure ongoing protein synthesis in starved cells. Starved cultures were amended with azidohomoalanine (AHA) for various labeling periods before cells were fixed and a fluorophore was covalently linked to incorporated AHA by strain-promoted azide-alkyne cycloaddition. Azidohomoalanine does not appear to be catabolized by *P. aeruginosa* and was unable to alleviate carbon or nitrogen limitation to permit growth (Fig. S1B and C). Although the average signal per starved cell was much lower than that observed for growing cells (Fig. S1F and H), it increased approximately linearly with labeling time in both starvation conditions (Fig. 1E and F). Fluorescence distributions did not show evidence of distinct sub-populations. Although AHA incorporation is not an absolute measure of protein synthesis, nitrogen-starved cells consistently showed higher signal per labeling time than carbon-starved cells (Fig. 1G and H), indicating relatively higher rates of new protein synthesis.

The ribosome content of growth-arrested *P. aeruginosa* was then probed under both starvation conditions using fluorescence *in situ* hybridization (FISH) (6, 32). Starved cells showed significantly lower signal than growing cells (Fig. S1E and G), but there was a trend (not statistically significant) toward nitrogen-starved cells having lower signal on average than carbon-starved cells despite the BONCAT indications of higher protein synthesis rates (Fig. 1I). These data indicated that distinct and dynamic translational strategies accompany the responses to the two starvation conditions and that ribosome content and protein synthetic rate are not directly correlated in growth-arrested *P. aeruginosa*.

## Starved *P. aeruginosa* undergoes bursts of translation during a starvation transition

Having defined two growth-arrested states with distinct translational activities, we next investigated how starved *P. aeruginosa* reacted to rapid transitions between the two. We collected nitrogen-starved cells by centrifugation and resuspended them in carbon starvation medium, hypothesizing that quickly shifting between starvation conditions would force cells to adapt their activities despite ongoing growth-arresting nutrient limitation. No substantial increase in CFU counts or culture absorbance followed the transition from nitrogen starvation to carbon starvation media, indicating that the media switch did not leave sufficient residual nutrients to support growth (Fig. S2A and B).

Two-hour BONCAT labeling pulses were used to compare new protein synthesis before, during, and after the transition from nitrogen starvation to carbon starvation media (Fig. 2A) and a control transition into fresh nitrogen starvation conditions (Fig. 2D). A large burst of translation accompanied the transition of cells from nitrogen starvation to carbon starvation conditions, with the average BONCAT signal increasing threefold immediately following the transition and subsequently decreasing (Fig. 2B and C). Within a day of the transition, the average BONCAT signal decreased to the low levels previously observed during carbon starvation (Fig. 1G and H). These translational bursts were also observed in transitions from carbon to nitrogen starvation medium (Fig. S2F through H) although, in this case, an upshift in protein biosynthesis could be predicted by the

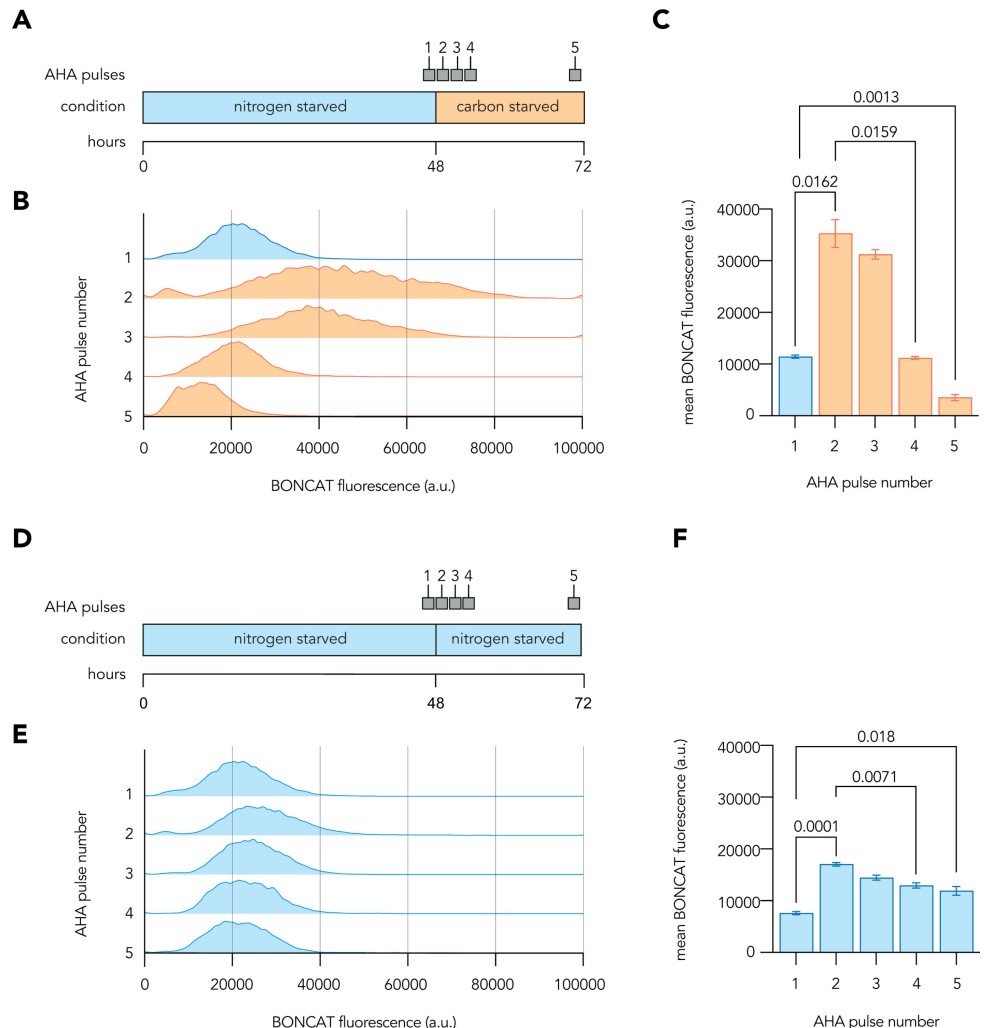

**FIG 2** Starved *P. aeruginosa* undergoes bursts of translation during a nutritional transition. (A and D) Schematics show the time-course of 2-h 500 µM AHA pulses applied to *P. aeruginosa* cultures for BONCAT labeling during a transition from nitrogen starvation to carbon starvation (A) and the transition from nitrogen starvation to new nitrogen starvation minimal media (D). (B and E) Histograms show representative distributions of per-cell fluorescence values from fixed BONCAT samples labeled across a transition between nitrogen and carbon, or nitrogen back to nitrogen, starvation. (C and F) Bar graphs show the mean BONCAT fluorescence values retrieved for each AHA labeling pulse. All data represent the mean value calculated from three biological replicates with error bars displaying standard deviation. One-way analysis of variance was conducted to assess differences in BONCAT signal across the labeling periods, with Dunnett's T3 multiple comparisons test used to calculate adjusted *P*-values for pairwise comparisons and selected significant values of interest shown. a.u. = arbitrary units.

higher average activity maintained during nitrogen starvation. Transitions from nitrogen starvation into fresh nitrogen starvation media elicited only a slight increase in average BONCAT signal (Fig. 2E and F), possibly due to replenishment of the carbon source, which should be consumed during ongoing nitrogen starvation. These data highlight that growth-arrested *P. aeruginosa* can quickly upregulate biosynthetic activity in response to environmental changes even during prolonged starvation, indicating that cells may maintain latent biosynthetic capacity during nutrient deprivation.

## The total and nascent proteomes of growth-arrested *P. aeruginosa*

We next investigated the portfolio of proteins present in and being produced by growth-arrested *P. aeruginosa* during nitrogen starvation, carbon starvation, and when

transitioning from nitrogen to carbon starvation. We first applied label-free quantification to the total proteome of *P. aeruginosa* by LC-MS/MS at several key points during a starvation time course including 48 h in nitrogen starvation, 24 h in carbon starvation, and a final outgrowth in nutrient-replete minimal medium allowing sampling during exponential phase (Fig. 3A). We normalized quantities of each protein to the total summed abundances of all proteins in each sample and identified more than 3,900 proteins with an average abundance of more than 1 part per million (ppm) in each condition. Principal component analysis demonstrated a tight clustering of replicate proteomes and highlighted that the proteomes of exponential phase *P. aeruginosa* cultures were distant from those of starved or transitioning cultures along principal component 1, which accounted for 61.1% of the variation (Fig. 3B). When the hundred most abundant proteins from total proteomes were compared across conditions we found substantial overlap, particularly among the most abundant proteins identified from starved and transitioning cultures, with 46 abundant proteins being shared among all nutrient-limited conditions (Fig. 3C; File S1). These abundant proteins were then functionally classified using a modified PseudoCAP (33) scheme (File S1) to allow for comparison of major cellular priorities under each condition. The abundant proteins of exponential phase proteomes were particularly enriched in translation-associated functions, while proteins relating to core cellular functions such as carbon compound anabolism and energy metabolism were abundant in all proteomes (Fig. S3A).

Relatively few distinctions were noted when comparing the abundance of individual proteins between the proteomes of starved and transitioning cultures, likely due to low amounts of new protein synthesis being overshadowed by the abundant proteome established at the onset of growth arrest (34) (Fig. S3B). Therefore, we employed BONCAT-linked proteomics to enrich and quantify the newly synthesized proteomes of *P. aeruginosa* under our conditions of interest. Cultures were starved of nitrogen for 48 h, the last six of which were labeled with AHA; then transitioned to carbon starvation, with a 3-h labeling period commencing immediately after the media switch, and a second labeling period of 12 h started after 24 h of carbon starvation (Fig. 3D). The labeling times were chosen to compensate for different rates of per-cell new protein synthesis observed in our previous fluorescence experiments (Fig. 2A through C). AHA-labeled proteins were enriched and analyzed by LC-MS/MS, as for the total proteomes. We identified more than 3,200 proteins with an average abundance of more than 1 ppm in each nascent proteome. Principal component analysis of the nascent proteomes revealed a tight clustering of replicates but suggested that the portfolios of nascent proteins made during different labeling periods were distinct (Fig. 3E). Indeed, the hundred most abundant proteins from each nascent proteome overlapped considerably less than the most abundant proteins in the total proteomes (Fig. 3F and C). Interestingly, we found similar patterns of functional enrichment among the nascent proteins from each condition, suggesting that although *P. aeruginosa* may express distinct proteins during each labeling period, similar general functions are maintained throughout the three labeling periods investigated (Fig. S3C).

When directly comparing nascent proteomes, we found that the abundance of hundreds of proteins differed substantially and significantly between the three labeling periods (Fig. 3G). Proteins from three functional categories of particular interest are highlighted: proteases and chaperones, flagellar motility, and the PTS[Ntr] system and its regulon (Fig. 3G; Fig. S3B; groups defined in File S1). These categories show relatively high expression, interesting dynamics, and/or functional importance in Tn-Seq data sets (as described below). As similar time-courses were used to generate the total and nascent proteomic data sets, we compared the relative abundances of individual proteins detected at analogous time points using scatter plots (Fig. 3H; File S1). In general, new protein synthesis reflects the composition of the total proteome, consistent with ongoing maintenance to broadly preserve cellular composition. However, points deviating from the diagonal reflect active proteome remodeling or proteins with unusually high or low turnover rates. Interestingly, many components of the PTS[Ntr]

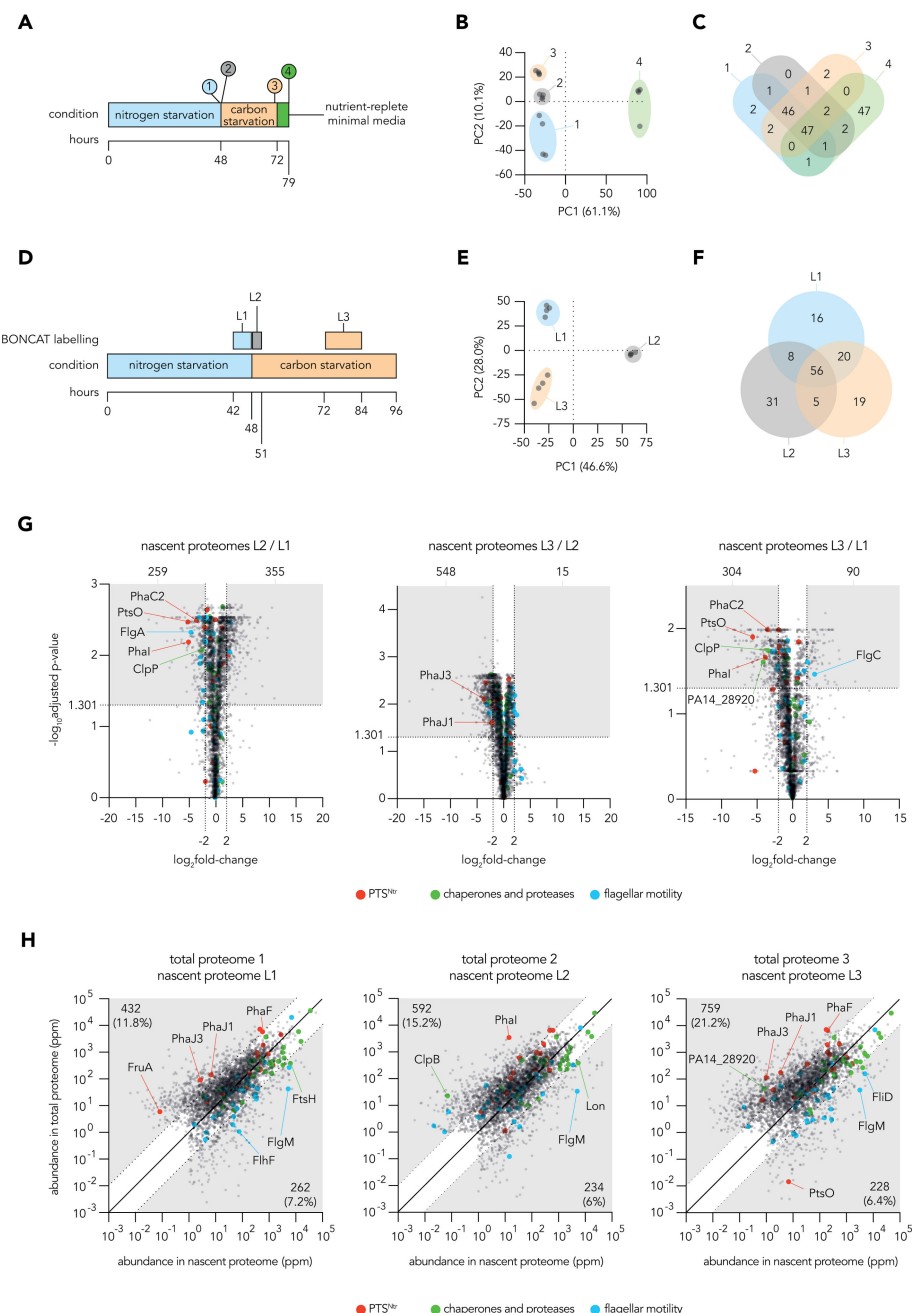

**FIG 3** The total and nascent proteomes of growth-arrested *P. aeruginosa*. (A and D) Schematics show the time-courses employed in total proteomic (A) and nascent (BONCAT) proteomic (D) experiments. (B and E) Scatter plots show principal component analyses of returned total (B) or BONCAT (E) proteomes. (C and F) Venn diagrams show the overlap of the top 100 most abundant proteins from each timepoint of total proteomic (C) and BONCAT proteomic (F) experiments. (G) Volcano plots show the fold change in abundance of nascent proteins when comparing different time points of the BONCAT proteomic experiment. In each volcano plot, shaded boxes highlight the nascent proteins which passed fold change ($\log_2$fold change > 2 or < −2) and Benjamini-Hochberg FDR-adjusted *P*-value ($\log_{10}$adjusted-*P*-value > 1.301) cutoffs to be considered as robust changes in abundance between compared proteomes, with the number of such proteins denoted above each shaded box. (H) Scatter plots combine total and nascent proteomic data and show the normalized average abundances of individual proteins detected in both data sets. Shaded regions indicate proteins that had greater than 10-fold difference in relative abundance between the two types of proteomic experiments, and the fraction of all detected proteins that fall within each shaded region is indicated. The larger colored dots highlight individual genes belonging to key physiological functions (full list of highlighted genes available in File S1, tab "functional categories"). All analyses used mean abundances calculated from four biological replicates.

network, such as proteins involved in the synthesis of polyhydroxyalkanoate (PHA) granules (35), have higher relative abundance in the total proteomic data, perhaps consistent with expression during the entry to growth arrest. Many flagellar proteins, proteases, and chaperones are represented more strongly in the nascent proteomes, indicating ongoing enhancement of their levels or high turnover. Overall, these data provide a detailed and comprehensive catalog of the total and nascent proteins in starved *P. aeruginosa*.

## The fitness determinants of growth-arrested *P. aeruginosa*

We next asked which genetic factors contributed to survival during our starvation conditions and across transitions between them. We utilized a high-density transposon mutant library to perform transposon-insertion sequencing and identify mutants that significantly changed in abundance after exposure to nitrogen or carbon starvation conditions. We also investigated how abundances changed following a transition to a second starvation condition (Fig. 4A). For each starvation sample collected, a short (~2 h, three doublings) outgrowth in LB medium was performed to allow starvation-imposed fitness defects to be amplified.

While many genes affected fitness during the first starvation exposure, only a small number of genes specifically impacted fitness across a starvation transition (Fig. 4B; File S1). Interestingly, many genes impacted fitness similarly in both starvation conditions (Fig. 4C and D), hinting that conserved processes and pathways generally support viability during growth arrest. Many genes highlighted in the nascent proteomic data robustly decreased in transposon read counts during carbon and/or nitrogen starvation, suggesting that expressing these proteins was beneficial for *P. aeruginosa* survival during prolonged growth arrest (Fig. 4E; File S1). All genes whose transposon read counts significantly and substantially changed during either starvation condition were then functionally categorized (Fig. 4F). We found that survival-enhancing and survival-diminishing genes identified from analyses of either starvation condition were similarly distributed across functional categories.

Interestingly, many genes encoding flagellar components increased in transposon read counts specifically following the transition of nitrogen-starved cultures to carbon starvation conditions (Fig. 4E). These data suggest that flagellar motility is costly, and while investing in exploration may have benefits in natural environments, loss of motility in the laboratory can improve fitness of growth-arrested *P. aeruginosa* navigating this transition. Overall, transposon insertion sequencing provided a holistic view of the fitness determinants impacting *P. aeruginosa* survival during starvation. In combination with corresponding proteomics data sets, these fitness data permit the identification of pathways and processes which are that prioritized and impactful during dynamic growth arrest.

## Patterns of expression and fitness impacts for key physiological functions

Having generated proteomic and TnSeq data sets from growth-arrested *P. aeruginosa*, we next concentrated on three functional categories that were generally highly expressed and important for fitness: flagellar motility, proteases and chaperones, and the PTS$^{Ntr}$ phosphotransferase system (Fig. S4A). We inspected the nascent proteome abundances and Tn-Seq read counts of individual components to analyze functional relationships within these categories (Fig. 5).

Collectively, flagellar proteins constitute roughly 2% of the nascent protein retrieved from all three BONCAT labeling timepoints although the flagellin protein FliC always ranked among the top 25 most abundant individual nascent proteins (Fig. S4B). Nascent flagellar proteins are most abundant during nitrogen starvation conditions and drop in relative abundance in the nascent proteomes of transitioning and carbon starved cells (Fig. 5A). Some highly abundant flagellar proteins and regulators (FliC, FliD, FlgM, FleN, and FlgG) were expressed during all three labeling timepoints, but the expression of many other flagellar components was dynamic. In general, proteins expressed earlier in

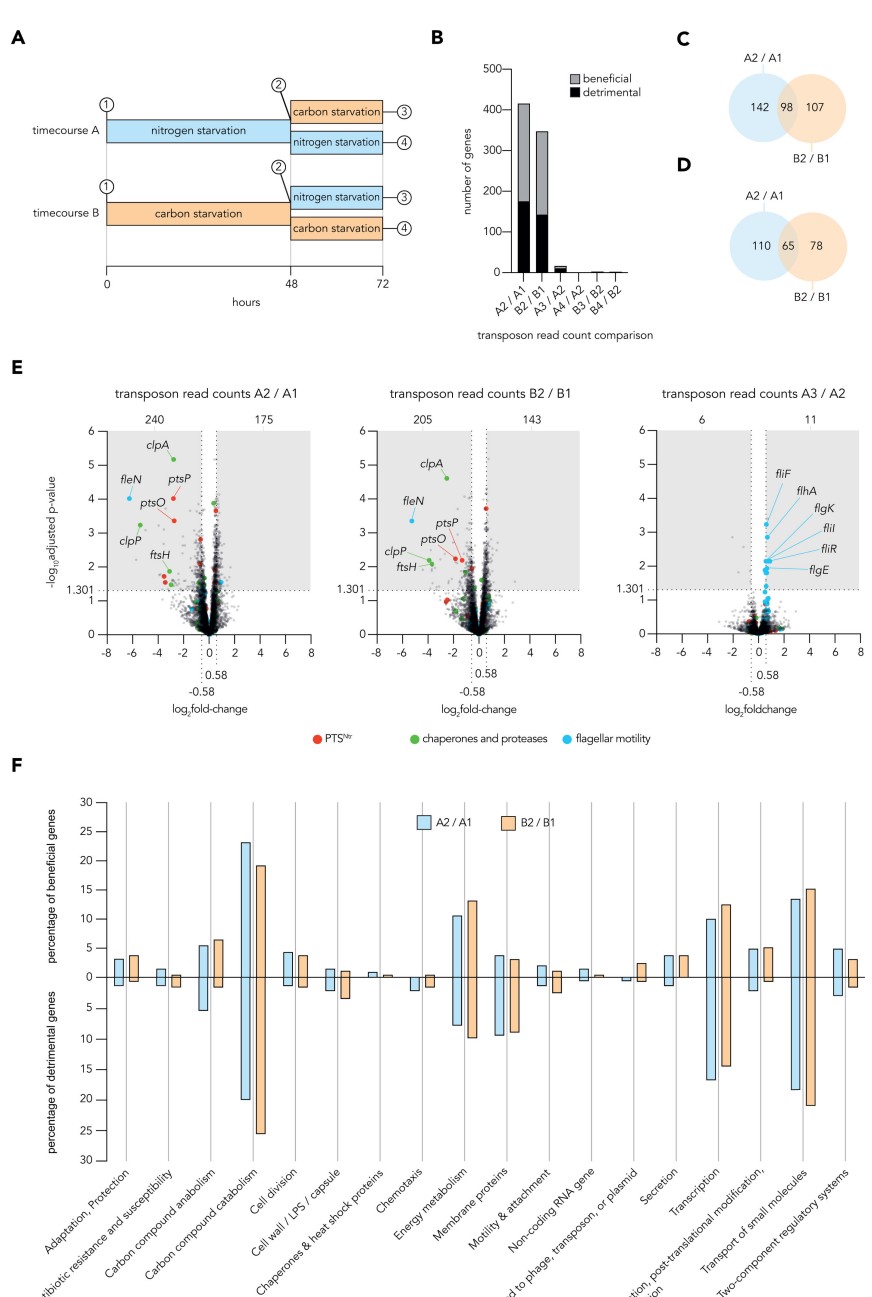

**FIG 4** The fitness determinants of growth-arrested *P. aeruginosa*. (A)Schematic shows TnSeq experimental time courses, highlighting the timepoints sampled and used in subsequent analyses. (B) Bar graph shows the numbers of beneficial genes whose transposon read count robustly decreased and detrimental genes whose transposon read count robustly increased in comparisons of library read counts between sampled timepoints. Fitness changes were considered robust if read counts crossed fold change (log$_2$fold change > 0.58 or < −0.58) and Benjamini-Hochberg FDR-adjusted *P*-value (log$_{10}$adjusted-*P*-value > 1.301) cutoffs when comparing samples. (C and D) Venn diagrams show the overlap of robust beneficial genes (C) and robust detrimental genes (D) in comparisons relevant to survival during nitrogen starvation (left, blue) and carbon starvation (right, orange). (E) Volcano plots show the fold change in transposon read counts and adjusted *P*-values for genes when comparing different sampling periods. In each plot, shaded boxes highlight the genes whose read count passed fold change and adjusted *P*-value cutoffs to be considered robust, with the number of these robust hits denoted above each shaded box. The larger colored dots highlight individual genes belonging to key physiological functions (full list of

Fig 4 (Continued)

highlighted genes available in File S1, tab "functional categories"). Some of these genes are obscured in some plots; only robust hits are annotated in each plot. (F) Bar graphs show the percentage occupancy of modified PseudoCAP groups by robust beneficial genes (top) and robust detrimental genes (bottom) retrieved from the comparisons of library read counts for nitrogen-starved or carbon-starved samples. Analyses used mean read counts calculated from six biological replicates with the exception of conditions A4 (nitrogen starvation back to nitrogen starvation) and B3 (carbon starvation to nitrogen starvation), which used five biological replicates.

flagellar biosynthesis, as part of the "class II" set transcriptionally regulated by FleQ, were more likely to have their highest relative expression during nitrogen starvation, while some members of the later classes "III/IV" regulated by FleR and FliA had notable bursts of new synthesis after the switch to carbon starvation (36). During individual starvations, read counts for most flagellar structural genes remain stable, implying flagellar defects had minimal impact to fitness during stable growth-arrested conditions. In contrast, read counts from genes encoding negative transcriptional regulators of flagellar biosynthesis, *fleN* and *flgM*, sharply decreased during both starvations, suggesting a high fitness cost of dysregulated overexpression of flagellar components. Together with the increased read counts for flagellar structural genes observed during starvation transitions, these data suggest that growth arrested *P. aeruginosa* carefully regulate ongoing flagellar activity. Nevertheless, proteomics data indicate that new synthesis of flagellar components remains a priority throughout prolonged starvation for *P. aeruginosa*.

Proteases and chaperones were extremely abundant in the nascent proteomes of growth-arrested *P. aeruginosa*, with this group constituting roughly 10% of nascent proteins at each labeled timepoint (Fig. S4B). Several individual proteases and chaperones, including GroEL, GroES, DnaK, FtsH, and Lon, were constitutively expressed and among the most abundant nascent proteins retrieved from all three labeling periods (Fig. 3 and 5B). Many non-essential protease and chaperone genes, such as *clpA*, *clpX*, *clpP*, and *ftsH*, exhibited strong and significant decreases in their read counts during starvation, suggesting components of this group are crucial for survival of growth arrest. Although read count increases were also observed in some proteases (e.g., *hslUV, dnaK*), these were small in magnitude and failed to pass thresholds of significance (File S1). For most proteases and chaperones, similar fitness impacts were observed regardless of the starvation condition, suggesting that they globally and generally contribute to starvation survival.

Transposon mutants of *ptsP* and *ptsO*, the first two components of the PTS$^{Ntr}$ phosphorylation cascade, had some of the strongest and most significant survival defects of any genes during both carbon and nitrogen starvations (Fig. 4E and 5C; File S1). This led us to interrogate the complex and enigmatic regulatory network surrounding the PTS$^{Ntr}$ in *P. aeruginosa*. PTS$^{Ntr}$ is highly conserved and modulates fluxes of carbon and nitrogen through bacterial metabolism via phosphorylation of PtsN, which has been proposed to regulate activity of diverse interaction partners in its unphosphorylated state (37–39). The phosphate derives from phosphoenolpyruvate, a central carbon metabolite, and PtsP activity is affected by 2-ketoglutarate and glutamine, providing a connection to nitrogen metabolism (40) (Fig. 5C). We interrogated our expression and fitness data for components of the PTS$^{Ntr}$ and fructose-related PTS systems; for metabolic enzymes impacting phosphoenolpyruvate, 2-ketoglutarate, and glutamine; and for known regulators of PHA metabolism, which is reported to be impacted by PTS$^{Ntr}$ in Pseudomonads (41). Transposon sequencing data suggested that in addition to *ptsP* and *ptsO,* genes required for phosphoenolpyruvate synthesis (*eno* and *pgm*) were important for survival during both starvation conditions (Fig. 5C). PTS$^{Ntr}$-related proteins constitute less than 1% of the nascent protein recovered from BONCAT labeling at each timepoint (Fig. S4B). However, some of the most highly produced proteins in this group (PtsP, PhaI, PhaC1) showed dynamic changes to expression, with strong decreases in abundance during the transition from nitrogen starvation to carbon starvation (Fig. 3G and 5C). In

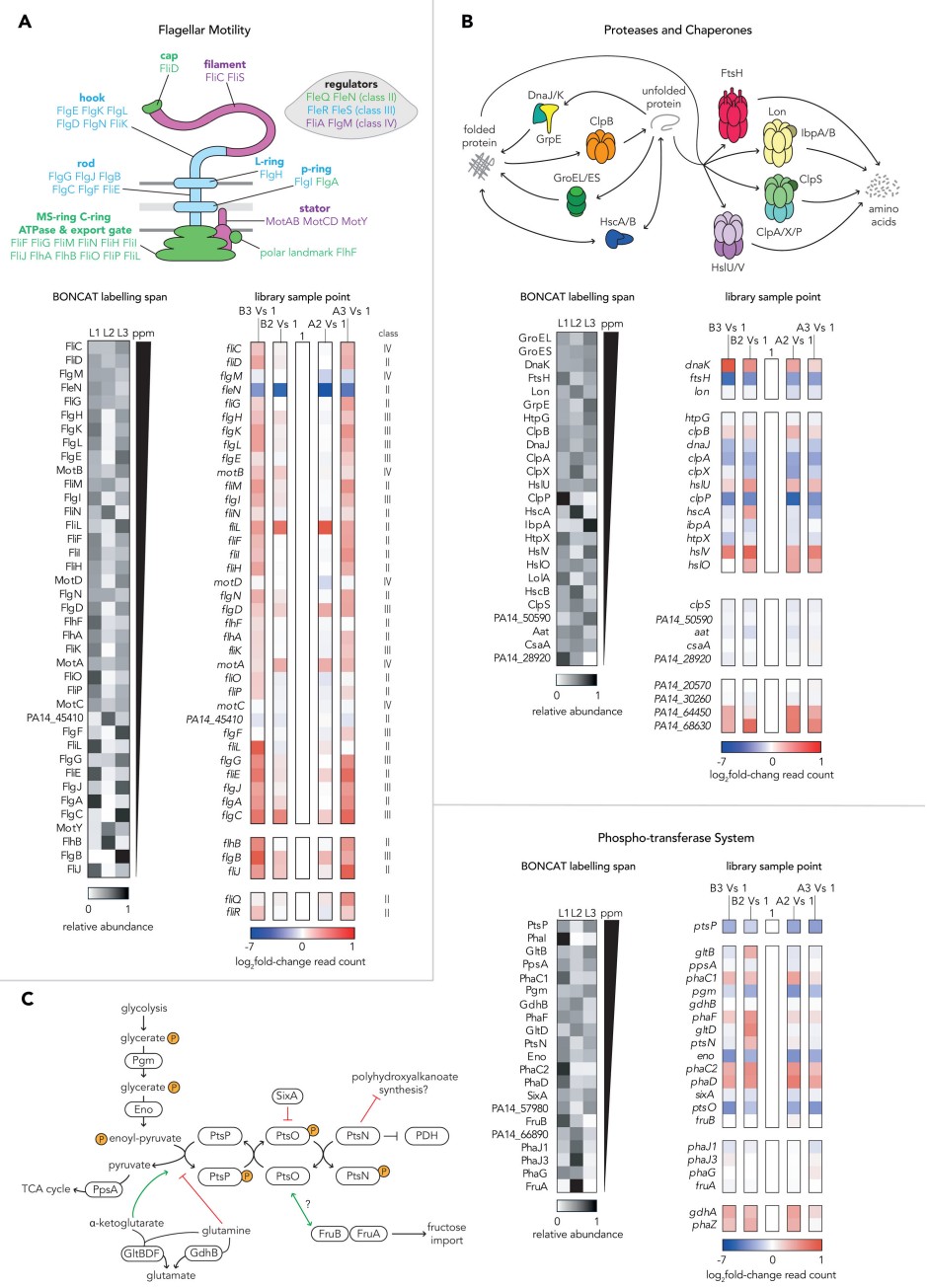

**FIG 5** Patterns of expression and read counts for key physiological functions. Schematics portray the structural or metabolic interplay of components belonging to each of the three key physiological functions highlighted by proteomic and TnSeq data. Genes and proteins of interest are grouped into three key pathways: flagellar motility (A), proteases and chaperones (B), and the phospho-transferase system (C). Black and white heat maps portray the fraction of total abundance, summed across the three labeling periods of the BONCAT proteomics experiment, associated with each time point. In each map, nascent proteins are ordered top to bottom by the total abundance summed across all three labeling periods. Colored heat maps portray the $\log_2$fold change in transposon read counts of genes of interest when comparing sampling timepoints of the transposon-insertion sequencing experiment. Each comparison of starved (A2 and B2) or transitioned (A3 and B3) read counts is made relative to stationary phase read counts (A1 and B1). Gaps in heat maps indicate missing data where nascent protein was not detected or a gene was absent from the transposon-mutant library. All analyses of BONCAT proteomic abundances used the mean abundances calculated from four biological replicates and analyses of TnSeq read counts used the means calculated from six biological replicates, with the exception of condition B3 (carbon starvation followed by nitrogen starvation), where only five replicates were used.

contrast, proteins with putative roles in the breakdown of PHA (PhaJ1, PhaJ3) increased during the transition.

## Validating impacts of three key functions in growth-arrested *P. aeruginosa*

Finally, we returned to our measures of per-cell size, ribosome abundance, and protein synthesis to investigate whether mutations in key representative genes identified from global proteomic and TnSeq data sets impacted these measures of growth arrested physiology. Based on observations from our global data sets that PTS-regulated processes were important, we also investigated PHA production using BODIPY[493/503] staining (42).

To inspect the role of proteases and chaperones during growth arrest, we focused on *ftsH* as a representative protease (22). Using a previously described (and genome-sequence-verified) clean deletion strain (43), we found that Δ*ftsH* cells had severely dysregulated new protein synthesis, producing significantly more protein during nitrogen starvation and significantly less during carbon starvation relative to wild-type cells (Fig. 6A). Concurrently, these Δ*ftsH* cells tended to possess more ribosomes in nitrogen starvation and fewer ribosomes in carbon starvation relative to wild-type cells, although these differences were not statistically significant (Fig. 6B), and they failed to produce a burst of protein synthesis when transitioned between starvation media (Fig. 6C). These data complement previous findings that *ftsH* contributes to fitness during growth arrest in *P. aeruginosa* by degrading damaged proteins, thus limiting the accumulation of toxic aggregates (43). In the alternating carbon and nitrogen starvation conditions, we tested in a direct competition with the wild type, Δ*ftsH* showed a slightly lower competitive index, but the difference was not statistically significant (Fig. 6D). These data suggest that individual proteases can dramatically, differentially, and consequentially sculpt the biosynthetic capacity of growth-arrested *P. aeruginosa* during nitrogen and carbon starvation.

We used non-motile transposon insertion mutant *flgE::MAR2xT7* (44) to probe the influence of flagellar motility during growth arrest (Fig. 1D). Although no substantial changes to protein synthesis and ribosome counts were found during carbon or nitrogen starvation, *flgE::tn* cells had a significantly larger burst of protein synthesis when transitioned between starvation conditions (Fig. 6A through C). Interestingly, these cells also stored more PHA than wild-type cells during nitrogen starvation (Fig. 6E). Together, these observations suggest that mutational loss of the capacity for flagellar motility frees resources for other biosynthetic activities during starvation.

Deletion mutants Δ*ptsP* (22) and Δ*ptsN* were used to probe impacts of the PTS during growth arrest. While the growth of Δ*ptsP* in nutrient-rich conditions was similar to wild type, Δ*ptsN* exhibited a growth defect, suggesting that unphosphorylated PtsN may play important roles during growth, while its phosphorylation is required for fitness during growth arrest (Fig. S5A). Both mutants had slightly (though not statistically significantly) reduced protein synthesis relative to wild type during nitrogen and carbon starvation (Fig. 6A), with Δ*ptsP* cells also possessing significantly fewer ribosomes during both starvation conditions (Fig. 6B). This mutant also failed to produce a translational burst during transitions between starvation conditions and exhibited a strong fitness defect when competed against wild-type cells during transitions (Fig. 6C and D). Furthermore, Δ*ptsP* cells appeared to elongate and failed to accumulate PHA during nitrogen starvation, as reported previously for other organisms (41) (Fig. 6E; Fig. S5C). These data show that the PTS[Ntr] impacts morphology and biosynthetic capacity in starved *P. aeruginosa* and validate *ptsP* as a crucial regulator of physiology in these growth-arrested contexts.

Overall, we attributed significant defects to the disruption of individual components of each of the three key functional categories identified by proteomic and TnSeq data. Proteases and chaperones and the PTS[Ntr] appear to severely and pleiotropically impact the physiology of starved cells, while the disruption of flagellar motility was specifically consequential during transitions, as suggested by the TnSeq data (Fig. 5). Interestingly,

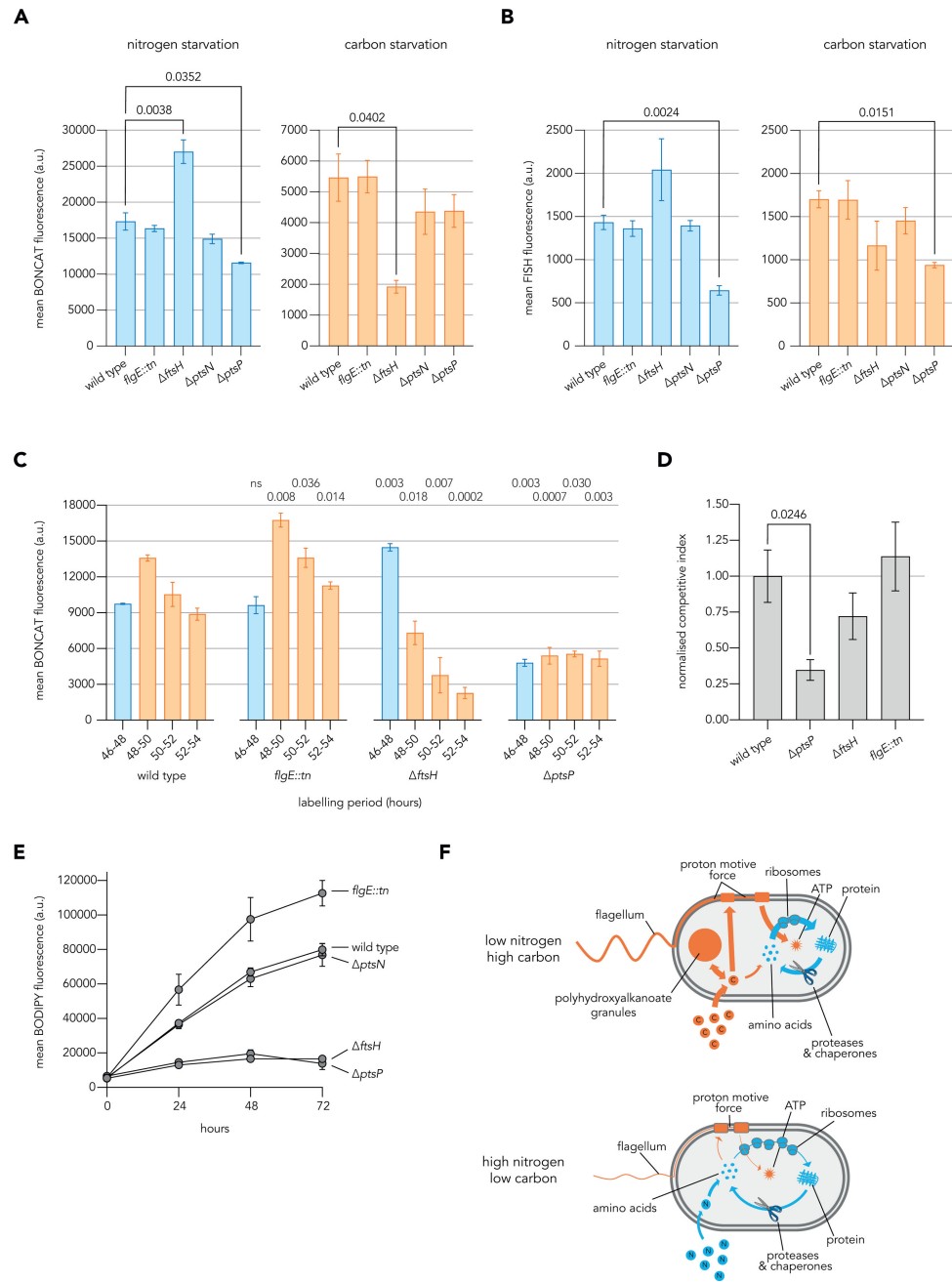

**FIG 6** Validating the impact of three key features in growth-arrested *P. aeruginosa*. (A) Bar graphs show the average cellular BONCAT fluorescence from the labeling of *P. aeruginosa* cultures for 240 min with 200 µM AHA following 48 h of incubation in nitrogen starvation (left) or carbon starvation (right) minimal media. (B) Bar graphs show the average cellular FISH fluorescence from the staining of *P. aeruginosa* cultures following 48 h of incubation in nitrogen starvation (left) or carbon starvation (right) minimal media. (C) Bar graphs show the average cellular fluorescence from pulsed BONCAT labeling of starved *P. aeruginosa* cultures during a transition between nitrogen starvation (blue) and carbon starvation (orange) minimal media. Cultures were incubated in nitrogen starvation minimal media for 48 h prior to the transition, with 2-h 1,000 µM AHA labeling pulses applied prior to (46–48) or following (48–50, 50–52, 52–54) the transition. Adjusted *P*-values for comparisons of each mutant labeling period to the analogous wild-type labeling period are shown above the bars. (D) Bar graph shows the normalized competitive index of strains competed against wild-type *P. aeruginosa* during three transitions between nitrogen starvation and carbon starvation minimal media. (E) Line graphs show PHA content as average per-cell BODIPY fluorescence of *P. aeruginosa* populations fixed at indicated timepoints during nitrogen starvation. (F) Schematics highlight the interplay of prioritized cellular processes and relevant resource stores in *P. aeruginosa* during growth arrest under different starvation

**Fig 6 (Continued)**

conditions. Arrow and line thickness represent the contribution of processes to the use of carbon or nitrogen sources in either starvation situation, as inferred from experimental data. All summarized data represent the mean values calculated from three biological replicates with error bars displaying standard deviation. One-way analysis of variance was conducted to assess differences between the wild-type and mutant strains, with Dunnett's T3 multiple comparisons test used to calculate adjusted *P*-values for pairwise comparisons and significant values shown. a.u. = arbitrary units.

defects in PHA accumulation and ribosome abundance were observed for mutants with strong fitness defects in both nitrogen and carbon starvation conditions.

## DISCUSSION

By combining quantitative single-cell techniques, BONCAT proteomics, and transposon-insertion sequencing, we probed the physiology of starved *P. aeruginosa* cells. Taken together, our results show that starvation-induced growth arrest is dynamic and actively regulated, with diverse new proteins continuously being synthesized for days by most cells in the population, albeit at much lower rates than during exponential growth. Our data offer insights into strategies that might support these dynamics. Growth arrest due to the absence of an essential macronutrient provides opportunities for storing other resources available in excess. Under fluctuating growth-arrested conditions, such internal stores can contribute to ongoing biosynthesis. Regulatory strategies must then determine when and how stored resources should be expended to meet challenges imposed by the environment. We propose three example intracellular resource storage mechanisms, relevant under different conditions of limiting and excess nutrients: ribosomes and other cellular proteins as storage for excess nitrogen, PHA granules as storage for excess carbon, and the proton motive force (PMF) as a store of energy (Fig. 6F). Resources can be shifted between these stores during starvation-induced growth arrest, and the three functional pathways we have highlighted (proteases and chaperones, flagella, and the PTS$^{Ntr}$ system) are involved in driving and regulating these shifts.

Ribosomes and proteins represent a major internal store of amino acids that could be liberated in times of need (45). During nitrogen starvation, where energy is available but amino acids are limiting for protein biosynthesis, it may be advantageous to more quickly recycle ribosomes and proteins to salvage nitrogen (45, 46) than during carbon starvation. Many bacterial species store ribosomes in inactive states during starvation by binding ribosome hibernation factors (20, 47, 48) and translation elongation can be slowed to different degrees in response to distinct nutrient limitations (21, 49). We observed evidence of slowed or hibernating ribosomes during carbon starvation, where cells showed relatively low protein synthesis, but high ribosome abundance compared to nitrogen starvation (Fig. 1H and I). Interestingly, the deletion of the protease *ftsH* leads to major dysregulation of ribosome abundance, new protein synthesis, and cell morphology during starvation (Fig. 6A through C; Fig. S5B). The increase in ribosome abundance during nitrogen starvation in Δ*ftsH* cells relative to the wild type is potentially consistent with a decrease in ribosome turnover. However, this mutant was also defective in maintaining ribosomes during carbon starvation, suggesting a more complex role of the protease.

We investigated PHA production, which represents a carbon storage mechanism, because defects in this function were previously reported for *ptsP* mutants in other organisms (41). Our Δ*ptsP* mutant was indeed defective in PHA storage (Fig. 6E), indicating that the PTS$^{Ntr}$ impacts carbon storage in *P. aeruginosa*. Surprisingly, we observed an increase in PHA synthesis in a mutant of flagellar hook component *flgE*. Powering flagella requires a sustained investment of energy derived from the PMF, which itself is maintained by respiration and ultimately carbon catabolism. The increased PHA content of non-motile cells suggests that PHA storage and PMF maintenance are linked during growth arrest, and hints that carbon flux between these anabolic and catabolic processes could represent an important buffer *P. aeruginosa* (Fig. 6F). Interestingly,

studies in *E. coli* have suggested direct links between the $PTS^{Ntr}$, flagellar motility, and resource storage (50, 51). The $PTS^{Ntr}$ system must also impact other aspects of cellular metabolism besides PHA storage, because mutations in *ptsP* and *ptsO* have strong fitness defects in multiple starvation conditions (22) (Fig. 5), while mutations in the PHA biosynthesis genes do not have strong effects under the starvation conditions we investigated. More work will be needed to gain a deeper understanding of how the $PTS^{Ntr}$ system influences resource distribution in *P. aeruginosa*, including carbon storage via PHA.

Flagellar biosynthesis and use appear to play prominent roles in the expenditure of cellular resources during starvation in *P. aeruginosa*. Starved cells were observed swimming (Fig. 1D), and our proteomics data suggest that flagellar synthesis continues during protracted starvation. An ongoing need for flagella is consistent with the recent report that *P. aeruginosa* continues swimming over long periods of carbon starvation, a tendency shared with a range of marine Gammaproteobacteria but not *E. coli* (31). However, the preservation of flagellar motility also appears to have a cost, as its dysregulation through mutation of *fleN* imposed one of the strongest fitness defects we observed in our TnSeq data. In contrast, mutations of many of the flagellar structural components conferred fitness advantages, specifically across the transition between nitrogen and carbon starvation conditions. Presumably, some of these mutants still incurred costs of partial flagellar biosynthesis, so these fitness impacts may relate more to expenditures of the PMF by a fully functioning flagellum. Interestingly, two other genes that showed significant fitness effect across the starvation transitions we investigated were inhibitors of efflux pump expression, *mexR* and *nfxB* (File S1). Mutations in these genes lead to the overexpression of efflux pumps, which deplete the PMF (52) and have previously been shown to cause loss of fitness under nutrient limitation (53). Flagellar motility and efflux pumps can confer obvious fitness benefits that justify their costs to the PMF, in the presence of toxins that must be removed from the cell or when better opportunities for colonization and growth exist at a distance. However, our data suggest that preserving PMF contributes substantially to fitness for *P. aeruginosa* in fluctuating growth arrested conditions (Fig. 6F).

Much more work will be required to investigate the details of molecular mechanisms orchestrating the dynamic changes in activity and resource distribution we have observed in starved *P. aeruginosa*. Our experiments represent reductionist "end cases" with well-mixed planktonic cultures totally starved for nitrogen or carbon. This approach was used to probe the possibilities for ongoing biosynthesis under nutritional extremes and to simplify the performance and interpretation of experiments. However, we propose that the same mechanisms are likely to contribute to buffering imbalances in resource availability in a range of non-steady-state growth conditions, which likely include complex natural habitats such as biofilms and human infection contexts. We hope that our genome-scale proteomic and genetic fitness data can serve as a resource to support future investigations of growth arrest regulation and physiology in *P. aeruginosa*.

## MATERIALS AND METHODS

Please see supplemental materials and methods for additional details.

### Strains, media, and handling

MOPS minimal medium lacking carbon and nitrogen sources (50 mM MOPS, 40 mM NaCl, 4 mM $K_2HPO_4$, 2.0 mM KCl, 1.0 mM $MgSO_4$, 0.1 mM $CaCl_2$, 7.5 µM $FeCl_2·4H_2O$, 0.8 µM $CoCl_2·6H_2O$, 0.5 µM $MnCl2·4H_2O$, 0.5 µM $ZnCl_2$, 0.2 µM $Na_2MoO_4·2H_2O$, 0.1 µM $NiCl_2·6H_2O$, 0.1 µM $H_3BO_3$, 0.01 µM $CuSO_4·5H_2O$) was used as a foundation for all nitrogen and carbon starvation experiments. Forty-five millimolar sodium succinate or 30 mM ammonium chloride was added as carbon or nitrogen sources, respectively, as appropriate. *P. aeruginosa* UCBPP-PA14 and derived mutant strains (Supplemental

material, strain List; all mutant strains were confirmed to contain the intended mutation and no additional mutations by whole genome sequencing) were consistently grown to stationary phase (~24 h) in lysogeny broth (10 g/L tryptone, 5 g/L yeast extract, 10 g/L NaCl) at 37°C shaking prior to their resuspension in starvation media. Starved cultures were shaken and maintained at 37°C in all experiments.

To transition between starvations, cells were pelleted and washed at least once with pre-heated MOPS medium lacking carbon and nitrogen before resuspension in the new medium. These transfers could result in the loss of up to 50%–75% of cells from cultures during transitions (Fig. S2A and B).

## Microscopy and flow cytometry

All images and videos were acquired using a Teledyne 48 photometrics camera fitted to a Nikon Eclipse Ti2 microscope with a 60× phase contrast oil objective (Nikon—MRD31605) and Spectra light engine (Lumencor). All microscopy images were processed in ImageJ using the microbeJ plugin (54). Microscopy-based experiments which quantified cell area parameters involved data from 60 to 300 cells per biological replicate. For analyzing larger populations of cells, flow cytometry on a Novocyte (Agilent) flow cytometer was used. Cytometry data were imported and analyzed in FlowJo V10.8.1 (Becton Dickinson). Where required, fixed samples were stained with 1 µg/mL BODIPY (ThermoFisher) for 60 min at 37°C in the dark and/or 10 µg/mL DAPI. Please see the supplemental material for additional analysis details.

## Fluorescence *in situ* hybridization

Cellular ribosome content was assessed using fluorescence *in situ* hybridization as previously described (32), with minor modifications. Briefly, Cy5-labeled EUB338 16S rRNA or control "scramble" probes (Integrated DNA Technologies) were hybridized for 3 h at 46°C in hybridization buffer (900 mM NaCl, 20 mM Tris [pH 7.6], 0.01% SDS, 20% HiDi formamide, and 2 µM fluorescent probe) against cells that had been fixed in 4% paraformaldehyde, permeabilized with ice cold ethanol and washed three times in 0.9% NaCl. Samples were then washed twice in wash buffer (215 mM NaCl, 20 mM Tris [pH 7.6], 5 mM EDTA) for 15 min at 48°C shaking in the dark and resuspended in 0.9% NaCl for use in flow cytometry and microscopy. Scramble probe intensity from a single replicate per condition was subtracted from EUB338 probe intensity for all three replicates during analysis.

## Fluorescence-conjugated BONCAT labeling

For whole cell BONCAT labeling, aliquots of culture were amended with azidohomoalanine (AHA; Tocris Bioscience), with labeling times and AHA concentrations indicated in figure legends. An AHA-free control was used for each condition and time point to determine background signal, with the intensity of this control subtracted from replicate labeled samples in analyses. Following labeling, cultures were pelleted using a benchtop centrifuge and washed three times in 0.9% NaCl to remove residual azidohomoalanine. 0.9% NaCl was used to dissolve all subsequent reagents and used to wash cells after each step. Cells were fixed with 4% paraformaldehyde at room temperature for 30 min, permeabilized in 70% ice cold ethanol, incubated with 100 mM iodoacetamide at 46°C for 30 min in the dark to alkylate free cysteines, incubated with 25 µM DBCO-AF488 or DBCO-AF647 (Fig. 1H; Fig. S1H) (Jena Bioscience) at room temperature for 30 min in the dark, and were finally washed three times in 0.9% NaCl to remove residual dye for use in microscopy and flow cytometry measurements.

## Proteomics

For both proteomics experiments, cultures of UCBPP-PA14 were grown for 24 h at 37°C shaking in LB before being normalized to an optical density of 0.2 in 100–300 mL

nitrogen starvation MOPS minimal media and incubated/sampled at time points as indicated (Fig. 3).

## Total proteomics experiments

Five milliliter aliquots were flash frozen with liquid nitrogen and stored at −70°C. Pellets were lysed in 2% SDS, 100 mM Tris (pH 8.0), 0.2 µL/mL benzonase with cOmplete EDTA-free protease inhibitor (Sigma Aldrich), clarified by spinning at high speed in a benchtop centrifuge (13,000 × $g$ for 10 min) and normalized to a protein concentration of 150 µg/mL using lysis buffer. Samples were submitted to the University of Dundee Fingerprints Proteomics facility and were processed using S-Trap Micro columns (Protifi) where proteins were reduced, alkylated, and digested overnight at 37°C with 0.75 µg of trypsin per sample. A second digest was repeated for 6 h the following day. Digested peptides were resuspended in 1% formic acid (FA) and run on a Q-Exactive HF instrument (Thermo Scientific) coupled to a Dionex Ultimate 3000 HPLC system (Thermo Scientific) with LC buffers comprising of buffer A (0.1% FA) and buffer B (80% ACN, 0.1% FA) over a gradient lasting 185 min where the peptides were eluted from a 110 cm uPAC Neo C18 column (Thermo Scientific) at a flow rate of 300 nL/min. Raw data were acquired in positive and Data Independent Acquisition (DIA) mode. A scan cycle compromised a full MS scan with an $m/z$ range of 345–1,155, resolution of 60,000, automatic gain control (AGC) target $3 \times 10^6$ and a maximum injection time of 200 ms. MS scans were followed by DIA scans of dynamic window widths. DIA spectra were recorded with a first fixed mass of 200 $m/z$, resolution of 30,000, AGC target $3 \times 10^6$, and a maximum IT of 55 ms. Normalized collision energy was set to 25% with a default charge state set at 3. Data for both MS scan and MS/MS DIA scan events were acquired in profile mode. Thermo RAW files were analyzed using Spectronaut version 17 (Biognosys) using directDIA. Samples were searched against the UCBPP-PA14 proteome (Uniprot reference UP000000653) with variable modifications of Oxidation(M), Dioxidation(MW), Acetyl (Protein N-term), Deamidation (NQ) and Gln->Pyro Glu set and a fixed modification of Carbamidomethyl (C). For identification, a precursor (posterior error probability (PEP) cutoff of 10% and a protein PEP cutoff of 7.5% were set and a false discovery rate of 1% was used. Quantification was done using the protein LFQ method Quant 2.0 (SN standard) within Spectronaut.

## BONCAT proteomics experiments

One hundred milliliter aliquots were removed from cultures and had AHA added to a final concentration of 500 µM for the time indicated in Fig. 3. Following labeling, 200 µg/mL chloramphenicol was added to cultures to terminate protein synthesis before pelleting, washing in 0.9% NaCl and flash freezing. Cells were lysed as for total proteomics with the addition of 100 mM iodoacetamide and lysates normalized to equivalent protein concentrations. Urea buffer (8 M urea, 150 mM NaCl, 1× protease inhibitor) and 25 µL of DBCO agarose beads (Discovery Dyes) were added to normalized samples and rotated for 4 h at room temperature. Beads were collected by centrifugation at 1,000 rpm, washed with SDS wash buffer (0.8% SDS, 150 mM NaCl, 100 mM Tris, pH 8.0), and resuspended in SDS wash buffer with 5 mM dithiothreitol at room temperature for 30 min. Beads were then incubated in SDS wash buffer with 100 mM iodoacetamide at 50°C in the dark for 45 min. Beads were washed in SDS wash buffer and transferred to Poly-Prep chromatography columns (BIO-RAD). Columns were washed eight times with 5 mL SDS wash buffer, eight times with 5 mL urea buffer with 100 mM Tris (pH 8.0), and eight times with 5 mL 20% acetonitrile. Finally, beads were resuspended in 10% acetonitrile with 50 mM ammonium bicarbonate and submitted to the proteomics facility for analysis. Peptides were recovered from the beads and supernatant of the submitted samples by tryptic digest and analyzed by LC-MS/MS as for total proteomes.

## Tn-Seq

Transposon insertion sequencing experiments were performed on two separate occasions, each using three biological replicates exposed to identical conditions and sampling time-courses (Fig. 4A). For both experiments, aliquots of the UCBPP-PA14 transposon mutant library (supplemental materials and methods) were thawed and 50 µL was used to inoculate 50 mL of LB. Stationary phase samples (~2 mL) were taken after 24 h growth and pelleted and frozen. The remainder of each LB culture was then pelleted, washed, and resuspended to a final OD of 0.2 in 40 mL of either carbon starvation or nitrogen starvation MOPS minimal media. 0.5 OD units were collected after 48 h incubation, washed, and resuspended in 5 mL LB for outgrowth. Following 2 h of outgrowth, with cultures reaching an OD of ~0.6, cultures were pelleted and frozen. The remaining cultures were split in half, pelleted, washed thoroughly in MOPS minimal media lacking both carbon and nitrogen sources, and resuspended in 10 mL of either nitrogen starvation or carbon starvation MOPS minimal media, achieving an OD of ~0.05. Transitioned cultures were incubated for 24 h at 37°C shaking before sampling. The entire remaining cultures were pelleted and resuspended in 5 mL of LB for outgrowth. Following 2 h of outgrowth, with cultures reaching an OD of ~0.6, cultures were pelleted and frozen. Sequencing libraries were prepared as previously described (22). Samples were then normalized, pooled, and sequenced on the Illumina NextSeq2000 instrument with a P1 reagent kit. Approximately 2.2–4.6 million single-end 100 bp reads were obtained per sample. Raw FASTQ files were processed for analysis using the Tn-Seq Pre-Processor (TPP) tool from the Transit package (55), mapping reads to the UCBPP-PA14 genome (NC_008463.1). Read counts per gene were summarized using the FeatureCounts algorithm of the subread software package (56), discounting reads that mapped to the first and last 50 bp of the gene. Both independent experiments were combined so that each sampling timepoint is represented by at least five biological replicates (File S1). The Voom/Limma differential expression method implemented on the Degust server was used to assess statistical significance of differences in read counts (57).

## ACKNOWLEDGMENTS

We are grateful for the ΔptsP and ΔftsH strains gifted to us by Dianne Newman. We appreciate assistance from the University of Dundee Flow Cytometry and Cell Sorting Facility, Imaging Facility, and Fingerprints Proteomics Facility as well as the sequencing facility at the James Hutton Institute. We would like to thank Laurent Delavaine, Daniel Neill, Lisa Racki, and Dianne Newman for helpful feedback and discussions.

Funding for this work was provided by the Wellcome Trust Institutional Strategic Support Fund to M.B. (204816/Z/16/Z), a Wellcome Trust PhD Studentship to F.D.M. (218520/Z/19/Z); the UK Academy of Medical Sciences (Springboard Award [SBF005/1096] to M.B.); and the UK Research and Innovation Medical Research Council (Future Leaders Fellowship [MR/T041811/1] and [MR/Z000378/1] to M.B.).

## AUTHOR AFFILIATION

[1]Division of Molecular Microbiology, University of Dundee School of Life Sciences, Dundee, United Kingdom

## AUTHOR ORCIDs

Findlay D. Munro http://orcid.org/0000-0003-4164-1544

Megan Bergkessel http://orcid.org/0000-0002-4530-1224

## FUNDING

| Funder | Grant(s) | Author(s) |
|---|---|---|
| Wellcome Trust | 204816/Z/16/Z,218520/Z/19/Z | Findlay D. Munro |
| | | Megan Bergkessel |
| Academy of Medical Sciences | SBF005/1096 | Megan Bergkessel |
| Medical Research Council | MR/T041811/1, MR/Z000378/1 | Megan Bergkessel |

## AUTHOR CONTRIBUTIONS

Findlay D. Munro, Conceptualization, Data curation, Formal analysis, Funding acquisition, Investigation, Methodology, Resources, Supervision, Validation, Visualization, Writing – original draft, Writing – review and editing | Elize Ambulte, Investigation, Resources, Validation | Claudia M. Hemsley, Investigation, Methodology, Resources, Supervision, Validation | Megan Bergkessel, Conceptualization, Data curation, Formal analysis, Funding acquisition, Investigation, Methodology, Project administration, Resources, Supervision, Validation, Visualization, Writing – original draft, Writing – review and editing

## DATA AVAILABILITY

All data used in the analyses presented here have been submitted to appropriate repositories of EMBL-EBI and can be accessed via the Biostudies accession number S-BSST2231 (doi: 10.6019/S-BSST2231). This includes flow cytometry data (.fcs files in a zipped archive), microscopy images uploaded to the BioImage Archive (accession S-BIAD2333), Tn-Seq raw sequence files and read counts per gene uploaded to Array Express (accession E-MTAB-15727), and proteomics raw data files and search results available via ProteomeXchange (https://www.ebi.ac.uk/pride/) with identifiers PXD069186 (total proteomes) and PXD069190 (BONCAT proteomes).

## ADDITIONAL FILES

The following material is available online.

### Supplemental Material

**File S1 (mSystems01439-25-s0001.xlsx).** Proteomics and Tn-Seq data plus gene lists used in analysis.
**Supplemental material (mSystems01439-25-s0002.pdf).** Figures S1-S5 and additional methodological details.

### Open Peer Review

**PEER REVIEW HISTORY (review-history.pdf).** An accounting of the reviewer comments and feedback.

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
