## [Reviewer comments · mSystems]

***Pseudomonas aeruginosa* dynamically prioritizes motility and resource recycling during prolonged starvation**

Findlay Munro, Elize Ambulte, Claudia Hemsley, and Megan Bergkessel

Corresponding Author(s): Megan Bergkessel, University of Dundee School of Life Sciences

Review Timeline:

Submission Date:	October 10, 2025
Editorial Decision:	December 1, 2025
Revision Received:	January 24, 2026
Accepted:	February 12, 2026

Editor: Soumya Kannan

Reviewer(s): The reviewers have opted to remain anonymous.

Transaction Report:

DOI: <https://doi.org/10.1128/mSystems.01439-25>

Re: mSystems01439-25 (**Analyses of protein expression and genetic fitness determinants reveal dynamic pathways active in starved *Pseudomonas aeruginosa***)

Dear Dr. Megan Bergkessel:

Both reviewers noted the high quality and thorough nature of this study, and as such we will be happy to consider the manuscript for publication following minor revisions as suggested in the review comments below, which note several points for expanded discussion or clarification.

Revision Guidelines

Sincerely,
Soumya Kannan
Editor
mSystems

Reviewer #1 (Comments for the Author):

I read the manuscript written by Munro & coworkers with great interest. It provides a comprehensive study of *Pseudomonas aeruginosa* (PA) physiology during nitrogen or carbon growth arrest. This is an important topic as PA is an opportunistic

pathogen that is not only associated with human hosts, but also frequently found in aquatic environments where it must survive prolonged periods of nutrient deprivation. Therefore, survival of PA under these conditions will give us a better understanding on bacterial survival (also of other species), and potentially may teach us better how opportunistic pathogens transfer from environmental reservoirs to human hosts.

The study is carefully done and employs a host of modern techniques to probe the gene expression, proteome, and survival of PA during carbon and nitrogen limitation, as well as the transfer from one to the other limitations. These results are showing that, contrary to expectations, there is a lot of gene expression dynamics hidden in an essentially dying culture and that is of wide interest to the scientific community. Therefore, I recommend publication of this manuscript, provided the few concerns I have, listed below, are addressed.

Major points

1. As the authors express in their introduction, one of the reasons that starvation in bacteria is understudied are a number of technical and conceptual challenges. It seems that in the case of steady-state growth, cells reach a physiological state where history does not matter, and there seems to be just one major parameter (the growth rate) that can describe the bacterial physiology well, despite bacteria growing on different nutrients. There seems to be no such equivalent in starvation, e.g. steady-state death, that is history-independent. Therefore, it is expected that the conditions prior to starvation are relevant for the specific physiology during starvation. In this case, the cells are grown in undefined rich media and likely reach well into stationary phase, before they are rapidly starved by washing. I am not suggesting the authors explore many other growth and starvation conditions, but this should be at least described and commented on in general. Two specific points related to this: A. In principle, shifting from amino-acid growth to another carbon source sets another perturbation on top of the nitrogen starvation. There is a fundamental metabolic constraint in bacterial cells that effectively prevents cells from simultaneously perform glycolysis as well as gluconeogenesis, and switching between the two conditions has been described to cause huge lag phases because cells need to switch from the one condition to the other [Basan et al. *Nature* 584 470-474 (2020)]. In this study I expect this effect to be small as the N- medium contains an organic acid, but there are still expected changes associated with biosynthesis genes that were not required in rich media. B. Rich media does not have a buffer, while the starvation medium (MOPS) is strongly buffered. This excludes a scenario where cells for example self-acidify during N limitation.
2. The authors describe that during both carbon and nitrogen limitation, cells continue to invest resources in motility. This is interesting and not obvious. Although the phenomenon of bacterial motility during starvation has been described before, this study adds that this motility is not a mere continuation of usage of the motility machinery (e.g. rotating flagellar motors) that was expressed before starvation, but is actively synthesised. This is relevant especially considering reductive divisions, which require new flagellar synthesis for every new cell. For a proper comparison, it would be helpful to also report the motility of the cells before starvation (Fig 1D). For growing populations of single-flagellated bacteria, one does not expect the fraction of motile cells to be 100% as this would require to start synthesising the flagellum at the other pole, peritrichous bacteria do not have this problem [Honda et al. *PNAS* 119 (37) e2110342119]. Also, I could not find the procedures that were used to determine the fraction of motile cells.
3. Statistics: the precise number of replicates should be stated (instead of at least 3) and for any statistical test used, the test statistic must be adjusted for multiple comparisons. It seems that this adjustment is not performed in this study. In many cases the corrections are minor as the number of conditions is low, but in others this may be more significant. I also encountered at least one instance (L364) where the main text describes a change that is not statistically significant. Something can be subtle yet statistically significant, and some apparently large changes can be insignificant.

Minor points

L47. I agree with the authors of the importance, but it is also fair to say that many bacterial physiology studies have actually not taken great care in establishing steady-state growth, showing that there is room in this field to improve attention to the culturing conditions.

L89: It is not clear to me what is "theoretical" about identifying the key physiological objectives in starvation. "Conceptual" perhaps?

L122: In the case of carbon storage, this is clear (lipids). In the case of nitrogen storage, it is less clear how this works.

-Fig 1. How do the authors reconcile the decrease in individual cell area (implicating decrease in cell mass) with the lack of decrease in cell number and absorbance. The expectation is that the number of cells multiplied by their biomass approximates the total absorbance [Zheng et al. *Nat Micro* 5, 995-1001 (2020)].

-Why OD500 instead of OD600? Perhaps this is obvious.

-Fig 1B seems inconsistent with Fig 1SA and B? Perhaps the conditions in the caption of Fig S1 could be clearer described.

-Fig 1A seems inconsistent with Fig S2 (LHS).

-The OD increase during C+N- conditions is likely because of lipid storage. The authors could do an experiment with their *ftsH* or *P* mutant to show that the absorbance in the first hours after starvation is different.

-Fig 4E, it was not clear to me if the highlighted proteins are labeled in a single plot or if this was consistent (e.g. the absence of the flagellar labels in the middle plot, does this mean they are not there or are they simply not highlighted?).

Reviewer #2 (Comments for the Author):

The manuscript "Analyses of protein expression and genetic fitness determinants reveal dynamic pathways active in starved *Pseudomonas aeruginosa*" by Munro et al. is a comprehensive study of the physiology of *P. aeruginosa* as it transitions to growth arrest due to nitrogen and carbon starvation, and due to a switch between nitrogen and carbon starvation. The manuscript is well written and informative. The authors use multiple lines of evidence to characterize protein remodeling during limitation to carbon or nitrogen. Among the experiments are FISH analysis of ribosome abundance and BONCAT analysis to quantify new protein synthesis. The investigators used BONCAT-based proteomics to identify nascent proteins that are synthesized due to nutrient starvation. They also performed TnSeq to identify genes required for optimal fitness during shifts to specific growth arrest. Finally, they used mutational studies on three pathways that they identified in the BONCAT and RNAseq analyses (flagella synthesis, proteases and chaperones, and the PTS uptake system) to gain information on the role of these pathways during transition to growth arrest. The discussion is nicely written, providing information on why these pathways are important.

Overall, this is an important study, demonstrating that new proteins are synthesized during growth arrest, and that the proteins that are produced may be specific for the type of nutrient starvation.

I have some minor comments that the authors may wish to address.

The investigators used MOPS buffer in their studies. It's apparent from their studies that *P. aeruginosa* can not grow on MOPS as a carbon and nitrogen source. Is it possible that *P. aeruginosa* can scavenge the nitrogen from MOPS. Perhaps the authors would like to comment on that.

On line 321, the authors describe *clpA*, *clpX*, *clpP*, and *ftsH* as non-essential. According to the Turner et al paper.

Proc Natl Acad Sci U S A 2015 Mar 31;112(13):4110-5. doi:10.1073/pnas.1419677112.

clpX, *clpP*, and *ftsH* are essential. Since the investigators were able to generate a *ftsH* mutant, that implies that it is not essential. The authors may like to elaborate on why their results differ from the Turner study. It might not be a bad idea to have your *ftsH* mutant sequenced, to head off any potential controversy about essential vs non-essential of that gene.

The fonts on the figures are pretty tiny. Can you increase the font size (at least on some of them), so that I don't have to get my reading glasses out as much?

This is just a suggestion: I find that titles that start with "Analysis of..." are kind of weak. This is a nice study. Please think about making the title stronger, focusing on what your primary results are, rather than the method used to find them.

The manuscript “Analyses of protein expression and genetic fitness determinants reveal dynamic pathways active in starved *Pseudomonas aeruginosa*” by Munro et al. is a comprehensive study of the physiology of *P. aeruginosa* as it transitions to growth arrest due to nitrogen and carbon starvation, and due to a switch between nitrogen and carbon starvation. The manuscript is well written and informative. The authors use multiple lines of evidence to characterize protein remodeling during limitation to carbon or nitrogen. Among the experiments are FISH analysis of ribosome abundance and BONCAT analysis to quantify new protein synthesis. The investigators used BONCAT-based proteomics to identify nascent proteins that are synthesized due to nutrient starvation. They also performed TnSeq to identify genes required for optimal fitness during shifts to specific growth arrest. Finally, they used mutational studies on three pathways that they identified in the BONCAT and RNAseq analyses (flagella synthesis, proteases and chaperones, and the PTS uptake system) to gain information on the role of these pathways during transition to growth arrest. The discussion is nicely written, providing information on why these pathways are important.

Overall, this is an important study, demonstrating that new proteins are synthesized during growth arrest, and that the proteins that are produced may be specific for the type of nutrient starvation.

I have some minor comments that the authors may wish to address.

The investigators used MOPS buffer in their studies. It's apparent from their studies that *P. aeruginosa* can not grow on MOPS as a carbon and nitrogen source. Is it possible that *P. aeruginosa* can scavenge the nitrogen from MOPS. Perhaps the authors would like to comment on that.

On line 321, the authors describe *clpA*, *clpX*, *clpP*, and *ftsH* as non-essential. According to the Turner et al paper.

Proc Natl Acad Sci U S A 2015 Mar 31;112(13):4110-5. doi:10.1073/pnas.1419677112.

clpX, *clpP*, and *ftsH* are essential. Since the investigators were able to generate a *ftsH* mutant, that implies that it is not essential. The authors may like to elaborate on why their results differ from the Turner study. It might not be a bad idea to have your *ftsH* mutant sequenced, to head off any potential controversy about essential vs non-essential of that gene.

The fonts on the figures are pretty tiny. Can you increase the font size (at least on some of them), so that I don't have to get my reading glasses out as much?

This is just a suggestion: I find that titles that start with “Analysis of...” are kind of weak. This is a nice study. Please think about making the title stronger, focusing on what your primary results are, rather than the method used to find them.

Reviewer #1 (Comments for the Author):

I read the manuscript written by Munro & coworkers with great interest. It provides a comprehensive study of *Pseudomonas aeruginosa* (PA) physiology during nitrogen or carbon growth arrest. This is an important topic as PA is an opportunistic pathogen that is not only associated with human hosts, but also frequently found in aquatic environments where it must survive prolonged periods of nutrient deprivation. Therefore, survival of PA under these conditions will give us a better understanding on bacterial survival (also of other species), and potentially may teach us better how opportunistic pathogens transfer from environmental reservoirs to human hosts.

The study is carefully done and employs a host of modern techniques to probe the gene expression, proteome, and survival of PA during carbon and nitrogen limitation, as well as the transfer from one to the other limitations. These results are showing that, contrary to expectations, there is a lot of gene expression dynamics hidden in an essentially dying culture and that is of wide interest to the scientific community. Therefore, I recommend publication of this manuscript, provided the few concerns I have, listed below, are addressed.

We appreciate the reviewer's interest in our work and the effort devoted to a very detailed and thorough review. The suggestions have greatly improved the manuscript.

Major points

1. As the authors express in their introduction, one of the reasons that starvation in bacteria is understudied are a number of technical and conceptual challenges. It seems that in the case of steady-state growth, cells reach a physiological state where history does not matter, and there seems to be just one major parameter (the growth rate) that can describe the bacterial physiology well, despite bacteria growing on different nutrients. There seems to be no such equivalent in starvation, e.g. steady-state death, that is history-independent. Therefore, it is expected that the conditions prior to starvation are relevant for the specific physiology during starvation. In this case, the cells are grown in undefined rich media and likely reach well into stationary phase, before they are rapidly starved by washing. I am not suggesting the authors explore many other growth and starvation conditions, but this should be at least described and commented on in general. Two specific points related to this: A. In principle, shifting from amino-acid growth to another carbon source sets another perturbation on top of the nitrogen starvation. There is a fundamental metabolic constraint in bacterial cells that effectively prevents cells from simultaneously perform glycolysis as well as gluconeogenesis, and switching between the two conditions has been described to cause huge lag phases because cells need to switch from the one condition to the other [Basan et al. Nature 584 470-474 (2020)]. In this study I expect this effect to be small as the N- medium contains an organic acid, but there are still expected changes associated with biosynthesis genes that were not required in rich media. B. Rich media does not have a buffer, while the starvation medium (MOPS) is strongly buffered. This excludes a scenario where cells for example self-acidify during N limitation.

We agree with these points and our only concern is the space in the manuscript that we can devote to them. We have now attempted to provide a bit more context. One nuance is that stationary phase cells are already not growing, so in our protocol at the point of the shift from (depleted) rich medium to the specific starvation states under investigation, there is not a sudden starvation; they are already starved for a limiting nutrient. Previously described growth lags observed when switching media between two growth-promoting conditions will not be relevant here because there is no measurable biomass growth in the starvation conditions of interest (although we agree that the referenced study into fluxes through central carbon metabolism is fascinating and important, and we also agree that our protocol should not require the bacteria to reverse flux through central carbon metabolism at any time. *P.*

aeruginosa prefers organic acids as a carbon source and should predominantly use them in all the conditions in our experiments).

We have chosen our starvation strategy to maximise starvation survival, because we want to study survival, not death, to the extent possible. We allow cells to slow their growth gradually while adapting to less favoured carbon sources (including fermentation byproducts) present in complex rich media, which has been shown by Schink and colleagues to promote starvation survival in *E. coli*^{1,2}. We then switch cells to buffered minimal media at lower density to avoid the subsequent pH stress and oxygen limitation that contribute to rapid death of a large fraction of the population in high-density LB stationary phase^{3,4}. In a previous publication we have discussed at length the potential physiological impacts of different models for inducing starvation and we now cite this publication⁵, along with the papers referenced above, while giving a brief explanation of our thinking (lines 108-114, line references throughout are to "compare" copy of manuscript). More generally, while our studies are undeniably reductionist, we hoped to choose conditions where we might gain some broadly relevant insights. In many natural contexts, heterotrophic bacteria colonise a niche with abundant nutrients, fill that niche by growing, and then disperse through a lower-nutrient environment while seeking a new nutrient-replete niche. We think that growing bacteria to stationary phase in LB before switching them to minimal media lacking a carbon or nitrogen source captures some features of this natural growth and dispersal cycle, and we acknowledge in our discussion the need for future work to more directly investigate whether dynamics similar to those we have described play out in more carefully replicated natural contexts (lines 532-537).

2. The authors describe that during both carbon and nitrogen limitation, cells continue to invest resources in motility. This is interesting and not obvious. Although the phenomenon of bacterial motility during starvation has been described before, this study adds that this motility is not a mere continuation of usage of the motility machinery (e.g. rotating flagellar motors) that was expressed before starvation, but is actively synthesised. This is relevant especially considering reductive divisions, which require new flagellar synthesis for every new cell. For a proper comparison, it would be helpful to also report the motility of the cells before starvation (Fig 1D). For growing populations of single-flagellated bacteria, one does not expect the fraction of motile cells to be 100% as this would require to start synthesising the flagellum at the other pole, peritrichous bacteria do not have this problem [Honda et al. PNAS 119 (37) e2110342119]. Also, I could not find the procedures that were used to determine the fraction of motile cells.

We agree that a comparison of starved cell motility to growing cell motility would be informative, so we have repeated the same measurements performed for starving cells on cells growing exponentially ($OD_{500}=0.2$) in LB. We have also now included a measurement of the average speeds of motile cells in all three conditions as well as the fraction of the population that is motile, for additional context. To fit these additions into the main figure, we have moved the control measurements performed on a flagellar mutant (*flgE::tn*) to supplemental figure 1. We have adjusted the text to reference the new data (lines 173-177, 866). As the reviewer has speculated, some cells are non-motile during exponential growth, and in fact a slightly (but not significantly) higher fraction of the population is motile during nitrogen starvation. To our knowledge, flagellar motility during total nitrogen starvation has not been extensively investigated in any Gammaproteobacteria but it is perhaps noteworthy that most of the genes in the flagellar synthesis cascade are transcribed by the nitrogen-starvation responsive sigma factor RpoN. Furthermore, running flagellar motors is energetically costly but nitrogen-starved cells could have excess energy from carbon catabolism that cannot be coupled to growth. This will be an interesting area for further investigation.

We apologise for the difficulty in finding the analysis details. They are included in the supplemental materials and methods. We have now added more detail to this section, and more explicitly directed to the supplemental materials and methods from the main text (line 574). Briefly, we used the Trackmate plug-in for ImageJ to track individual cells suspended in media on a coverslip and imaged by phase-contrast microscopy. We took 15-second timelapses at 3 frames per second and discarded tracks lasting less than 1.5 seconds. We set a threshold of 4 microns of total track displacement as the minimum for a cell to be considered motile, based on the behaviour of the *flgE::tn* mutant under the starvation conditions.

3. Statistics: the precise number of replicates should be stated (instead of at least 3) and for any statistical test used, the test statistic must be adjusted for multiple comparisons. It seems that this adjustment is not performed in this study. In many cases the corrections are minor as the number of conditions is low, but in others this may be more significant. I also encountered at least one instance (L364) where the main text describes a change that is not statistically significant. Something can be subtle yet statistically significant, and some apparently large changes can be insignificant.

We appreciate this careful attention to detail as it has caught an important error. Multiple comparison corrections were in fact applied in most statistical tests throughout this work, but they were not properly described in the figure legends and not consistently applied; in some cases p-values were overcorrected and in others they were not corrected. We have carefully re-assessed all statistical tests and corrected figure legends to precisely indicate the number of replicates and the statistical tests used (lines 876-880; lines 889-891; 919-920; 933; 944-946; 963-965; 975-976; 986-989). The Benjamini-Hochberg FDR control was applied to proteomics and Tn-Seq p-value calculations. For all other experiments, Brown-Forsythe/Welch one-way analysis of variance with post-hoc Dunnett's T3 multiple comparison tests were performed to calculate p-values for pairwise comparisons among multiple conditions, timepoints, or strains. This strategy was deemed most appropriate for our small sample sizes of 3 biological replicates. In the case of comparisons among strains, each mutant strain was compared only to the wild-type, and multiple

comparison corrections were made only for these comparisons. In some cases, previously reported p-values had been corrected for all possible comparisons among strains, but assessing differences among mutants was not part of the experimental design, and statistically significant differences between two different mutant strains were not reported or discussed, so it was inappropriate to correct for these (possible, but ignored) comparisons. We have also adjusted the text to clearly indicate when we have described a biologically interesting but not statistically significant trend (for example, lines 203-204; lines 406-408; line 413, line 441). We agree that it is important to clearly report the results of statistical tests, but in the context of subtle effects, small numbers of replicates, and exploration of a wider regulatory network, trends that do not meet a strict test for statistical significance can still be of interest. While revisiting the statistical analysis for all figures, we realised that the flow cytometry data shown in Figure 1G and Figure 1H had been collected on an older instrument than the data throughout the rest of the paper and presented substantially different absolute values for fluorescence. To avoid confusion and for more straightforward comparisons between figures, we have now replaced these data with data collected on the same instrument as the rest of the flow cytometry experiments in the paper. With these data and properly corrected p-values, the difference in ribosome abundance between carbon and nitrogen starvation is not significant, but it is a trend consistently observed in experiments throughout the paper. We apologise for our oversight in this area.

Minor points

L47. I agree with the authors of the importance, but it is also fair to say that many bacterial physiology studies have actually not taken great care in establishing steady-state growth, showing that there is room in this field to improve attention to the culturing conditions.

We completely agree with this and have cited a paper showing that steady state growth is only achieved in batch cultures of *E. coli* when a dilution of at least 1:10,000 is performed, and it ends at an OD₆₀₀ of approximately 0.1⁶. Most "exponential phase" experiments do not meet these conditions, and the populations being studied are likely not at steady state from a single cell perspective. We have made this point more explicitly (lines 65-67).

L89: It is not clear to me what is "theoretical" about identifying the key physiological objectives in starvation. "Conceptual" perhaps?
We agree that this is a better word and have adopted it (line 93).

L122: In the case of carbon storage, this is clear (lipids). In the case of nitrogen storage, it is less clear how this works.

This is an interesting question for future investigation, but it seems likely that inactive ribosomes (and unused proteins more generally) can serve an internal store of nitrogen during starvation. Ribosomes have well-characterised mechanisms by which they can enter inactive, stored states, and account for a substantial fraction of the cellular nitrogen in bacteria. We describe this possibility in the discussion, but we have now also briefly mentioned it in the introduction (lines 99-101).

-Fig 1. How do the authors reconcile the decrease in individual cell area (implicating

decrease in cell mass) with the lack of decrease in cell number and absorbance. The expectation is that the number of cells multiplied by their biomass approximates the total absorbance [Zheng et al. Nat Micro 5, 995-1001 (2020)].

This is an interesting point and may be another example where growth-arrested cells do not always exhibit relationships described for growing cells. We note that Zheng *et al.* focused exclusively on data from growing cultures (albeit growing at a wide range of rates). For example, growth-arrested cells will eventually begin to lose viability but may not concurrently lose optical density depending on their manner of death, leading to a disconnect between CFU numbers and absorbance. Changes in cell shape and composition during starvation could affect the relationship between the areas of fixed cells measured by phase-contrast microscopy and absorbance of the live culture. We have modified our description of these results to focus less on differences in the magnitudes of changes across these three parameters during carbon starvation and acknowledge that all three parameters decrease during carbon starvation at later timepoints (lines 155-162), but cell numbers and sizes are relatively stable in both conditions over the timepoints used throughout the manuscript, 24-96 hours after shifting to the starvation conditions (lines 177-180).

-Why OD500 instead of OD600? Perhaps this is obvious.

Some *P. aeruginosa* researchers (including us) measure OD at 500 nm to avoid interference from the phenazine pyocyanin, which has a broad absorbance peak from about 600-800 nm in aqueous solution. Pyocyanin production is highly variable depending on nutrient conditions and cell density. We have added a phrase to explain this (lines 119-120).

-Fig 1B seems inconsistent with Fig 1SA and B? Perhaps the conditions in the caption of Fig S1 could be clearer described.

We have added some additional clarification to both the main text and the supplemental figure legend (lines 151, 864-864 in main text). A key difference between the main text figure and the supplemental figures is that the supplemental figures show readings every 6 minutes for 10 hours, while the main text figure shows readings only every 2 hours over the first 10 hours, but then show readings at intervals for 200 hours.

-Fig 1A seems inconsistent with Fig S2 (LHS).

It is true that the error bars for the CFU determination in the first ~10 hours of nitrogen starvation are large in both figures, and the averages do not precisely match. We do routinely observe high variability in CFU counts over this timeframe. We would argue that the two figures are not inconsistent with each other, but both show substantial variability. Both experiments show some increase in CFUs during the initial hours of nitrogen starvation on average, but we agree the magnitude is larger for the main text figure. These are just the results obtained in two different executions of a long-term nitrogen starvation.

-The OD increase during C+N- conditions is likely because of lipid storage. The authors could do an experiment with their *ftsH* or *P* mutant to show that the absorbance in the first hours after starvation is different.

As we show in Supplemental Figure 5, both mutants have substantial morphological abnormalities including altered cell lengths during nitrogen starvation. These phenotypes could confound analysis of the impact of lipid storage on absorbance if

they affect absorbance independently of changes to lipid storage. However, we agree with the reviewer that the increase in OD during prolonged nitrogen starvation is primarily due to lipid storage, and we have investigated this in more detail for a different manuscript that is in preparation. We see that a mutant that is incapable of producing PHA granules has a smaller increase in OD during nitrogen starvation than the wild type, and a mutant that is incapable of depolymerising stored polyhydroxyalkanoate has a much larger increase in OD during nitrogen starvation. However, we think that these mutants and data are outside the scope of the current manuscript, which is already quite long.

-Fig 4E, it was not clear to me if the highlighted proteins are labeled in a single plot or if this was consistent (e.g. the absence of the flagellar labels in the middle plot, does this mean they are not there or are they simply not highlighted?).

The same genes are coloured in every plot, but they are only annotated if they stand out as robustly affected. We have clarified this in the figure legend and referenced the full list of highlighted genes, which is contained in supplemental file 1 (lines 926-927; 940-941). Highlighted gene dots are sometimes obscured by other gene dots in the mass of genes that do not have robust fitness effects.

Reviewer #2 (Comments for the Author):

The manuscript "Analyses of protein expression and genetic fitness determinants reveal dynamic pathways active in starved *Pseudomonas aeruginosa*" by Munro et al. is a comprehensive study of the physiology of *P. aeruginosa* as it transitions to growth arrest due to nitrogen and carbon starvation, and due to a switch between nitrogen and carbon starvation. The manuscript is well written and informative. The authors use multiple lines of evidence to characterize protein remodeling during limitation to carbon or nitrogen. Among the experiments are FISH analysis of ribosome abundance and BONCAT analysis to quantify new protein synthesis. The investigators used BONCAT-based proteomics to identify nascent proteins that are synthesized due to nutrient starvation. They also performed TnSeq to identify genes required for optimal fitness during shifts to specific growth arrest. Finally, they used mutational studies on three pathways that they identified in the BONCAT and RNAseq analyses (flagella synthesis, proteases and chaperones, and the PTS uptake system) to gain information on the role of these pathways during transition to growth arrest. The discussion is nicely written, providing information on why these pathways are important.

Overall, this is an important study, demonstrating that new proteins are synthesized during growth arrest, and that the proteins that are produced may be specific for the type of nutrient starvation.

We thank the reviewer for their interest in our work and for their suggestions to improve the manuscript.

I have some minor comments that the authors may wish to address.

The investigators used MOPS buffer in their studies. It's apparent from their studies

that *P. aeruginosa* can not grow on MOPS as a carbon and nitrogen source. Is it possible that *P. aeruginosa* can scavenge the nitrogen from MOPS. Perhaps the authors would like to comment on that.

We have seen no evidence in our work that *P. aeruginosa* can make any productive use of the nitrogen present in the MOPS buffer molecule. As the reviewer has noted, we showed that growth is not supported when either MOPS plus succinate or MOPS plus ammonium chloride are provided, but robust growth is observed in MOPS plus succinate and ammonium chloride. The MOPS is present at a concentration of 50 mM, so even if only a small percentage of it were able to be degraded and enter into amino acid biosynthetic pathways, it should be sufficient to support detectable growth. We chose to use the MOPS buffer because of previous reports that it is better than a phosphate buffered medium like M9 for supporting starvation survival in *E. coli*, perhaps because it imposes less osmotic stress. However, we have also performed BONCAT experiments in phosphate buffered minimal media without any MOPS present, and we have observed broadly similar behaviour in terms of new protein synthesis in nitrogen starvation conditions. If the nitrogen in MOPS is not able to support growth or protein synthesis, we do not know another way to detect that bacteria are scavenging it. We have clarified that this is our conclusion based on the growth data presented in figure 1 and supplemental figure 1 (lines 151-152).

On line 321, the authors describe *clpA*, *clpX*, *clpP*, and *ftsH* as non-essential. According to the Turner et al paper.

Proc Natl Acad Sci U S A 2015 Mar 31;112(13):4110-5.
doi:10.1073/pnas.1419677112.

clpX, *clpP*, and *ftsH* are essential. Since the investigators were able to generate a *ftsH* mutant, that implies that it is not essential. The authors may like to elaborate on why their results differ from the Turner study. It might not be a bad idea to have your *ftsH* mutant sequenced, to head off any potential controversy about essential vs non-essential of that gene.

We thank the reviewer for bringing this to our attention. We have already performed whole genome sequencing on all the mutant strains we investigated in this work, and we found no evidence for unexpected background mutations, plus we confirmed the presence of the expected mutations. This was noted only in the supplemental materials and methods where it is not obvious. We have now added mentions in the results section where we discuss the Δ *ftsH* mutant phenotypes (lines 402-403) and in the main text materials and methods (lines 556-558).

It is true that the Turner study found that *ftsH* met their criteria for essentiality in both strains of *P. aeruginosa* that they investigated, and additionally, this gene is described as essential in *E. coli*. Indeed, a gene essentiality analysis of our pooled transposon mutant library also calls *ftsH* as essential due to a statistically significant decrease in transposon insertion reads mapping to it relative to the expected density. One interesting observation is that *ftsH* is predicted to be the first gene in an operon that contains two more genes called as essential in analysis of Tn-Seq data (*folP* and *glnM*), so this may influence the competitive fitness of transposon insertion mutants in *ftsH*, even though the transposon contains a promoter intended to

diminish polar effects. Although we, Basta *et al.*, and Turner *et al.* found significantly fewer transposon insertions in this gene than expected, all studies detected some insertions. Also, there were enough reads in the starting stationary phase condition in our study to detect a strong and statistically significant reduction following starvation and outgrowth.

Clean deletions of *ftsH* in *P. aeruginosa* have been described in previous publications^{7,8}. Both studies reported complementation of the observed deletion phenotypes when they re-introduced the *ftsH* gene, another line of evidence that the strain is a *bona fide* deletion strain. The strain we have used in our work is the same one that was used in the Basta *et al.* publication, a gift from Dianne Newman. Both in our hands and as reported by Basta *et al.*, the Δ *ftsH* strain does have pleiotropic phenotypes and substantial fitness defects in multiple conditions. The severely reduced competitive fitness in transposon mutant libraries likely reflects this, but our and Basta *et al.*'s characterisation of the clean deletion suggest that the gene is not strictly essential in *P. aeruginosa* by a conventional genetic definition.

For the *clpX* and *clpP* genes, if I understand the supplemental data from the Turner *et al.* paper correctly, it seems that they were deemed to be conditionally essential only in the PAO1 strain and only in MOPS minimal media conditions. This is interesting given our results and suggests that much less stringent nutrient limitation than the total starvation we imposed might be sufficient to reveal a fitness defect for these protease genes.

The fonts on the figures are pretty tiny. Can you increase the font size (at least on some of them), so that I don't have to get my reading glasses out as much?

We apologise for the small font size and agree that the figures are improved by increasing it. We have increased the minimum size by at least one point in all figures.

This is just a suggestion: I find that titles that start with "Analysis of..." are kind of weak. This is a nice study. Please think about making the title stronger, focusing on what your primary results are, rather than the method used to find them.

We appreciate this suggestion and hope that our new title is more compelling:
***Pseudomonas aeruginosa* dynamically prioritizes motility and resource recycling during prolonged starvation**

References

- 1 Schink, S., Ammar, C., Chang, Y. F., Zimmer, R. & Basan, M. Analysis of proteome adaptation reveals a key role of the bacterial envelope in starvation survival. *Mol Syst Biol* **18**, e11160, doi:10.15252/msb.202211160 (2022).
- 2 Biselli, E., Schink, S. J. & Gerland, U. Slower growth of *Escherichia coli* leads to longer survival in carbon starvation due to a decrease in the maintenance rate. *Mol Syst Biol* **16**, e9478, doi:10.15252/msb.20209478 (2020).
- 3 Finkel, S. E. Long-term survival during stationary phase: evolution and the GASP phenotype. *Nat Rev Microbiol* **4**, 113–120, doi:10.1038/nrmicro1340 (2006).

- 4 Farrell, M. J. & Finkel, S. E. The growth advantage in stationary-phase phenotype conferred by *rpoS* mutations is dependent on the pH and nutrient environment. *J Bacteriol* **185**, 7044–7052 (2003).
- 5 Bergkessel, M. & Delavaine, L. Diversity in starvation survival strategies and outcomes among heterotrophic Proteobacteria. *Microb Physiol*, 1–17, doi:10.1159/000516215 (2021).
- 6 Roller, B. R. K. *et al.* Single-cell mass distributions reveal simple rules for achieving steady-state growth. *mBio* **14**, e01585–01523, doi:10.1128/mbio.01585-23 (2023).
- 7 Hinz, A., Lee, S., Jacoby, K. & Manoil, C. Membrane proteases and aminoglycoside antibiotic resistance. *J Bacteriol* **193**, 4790–4797, doi:10.1128/JB.05133-11 (2011).
- 8 Basta, D. W., Angeles-Albores, D., Spero, M. A., Ciemniecki, J. A. & Newman, D. K. Heat-shock proteases promote survival of *Pseudomonas aeruginosa* during growth arrest. *Proc Nat Acad Sci USA* **117**, 4358–4367, doi:10.1073/pnas.1912082117 (2020).

Re: mSystems01439-25R1 (*Pseudomonas aeruginosa* dynamically prioritizes motility and resource recycling during prolonged starvation)

Dear Dr. Megan Bergkessel:

Thank you for your patience and for the revised submission - we are happy to accept the manuscript as all reviewer comments have been sufficiently addressed. We hope to receive more of your work in mSystems in the future!

I am forwarding your manuscript to the ASM production staff for publication. Your paper will first be checked to make sure all elements meet the technical requirements. ASM staff will contact you if anything needs to be revised before copyediting and production can begin. Otherwise, you will be notified when your proofs are ready to be viewed.

Sincerely,
Soumya Kannan
Editor
mSystems